# Controlling gene expression with deep generative design of regulatory DNA

Jan Zrimec ●[1,2,8] ✉, Xiaozhi Fu ●[1,8], Azam Sheikh Muhammad ●[3], Christos Skrekas[1], Vykintas Jauniskis[1,4], Nora K. Speicher[3], Christoph S. Börlin[1,5], Vilhelm Verendel[3], Morteza Haghir Chehreghani[3], Devdatt Dubhashi[3], Verena Siewers[1], Florian David[1], Jens Nielsen ●[1,5] & Aleksej Zelezniak ●[1,6,7] ✉

Design of de novo synthetic regulatory DNA is a promising avenue to control gene expression in biotechnology and medicine. Using mutagenesis typically requires screening sizable random DNA libraries, which limits the designs to span merely a short section of the promoter and restricts their control of gene expression. Here, we prototype a deep learning strategy based on generative adversarial networks (GAN) by learning directly from genomic and transcriptomic data. Our ExpressionGAN can traverse the entire regulatory sequence-expression landscape in a gene-specific manner, generating regulatory DNA with prespecified target mRNA levels spanning the whole gene regulatory structure including coding and adjacent non-coding regions. Despite high sequence divergence from natural DNA, in vivo measurements show that 57% of the highly-expressed synthetic sequences surpass the expression levels of highly-expressed natural controls. This demonstrates the applicability and relevance of deep generative design to expand our knowledge and control of gene expression regulation in any desired organism, condition or tissue.

Gene expression is a fundamental process underlying the cellular functionality of all living organisms. Researchers have been trying to control it for decades, since it can help us design efficient gene therapies[1] and microbial cell factories[2], hopefully curing cancer among other diseases and aiding the transformation to a sustainable biobased society. Our ability to control gene expression derives from understanding the cell's intrinsic regulatory code[3], which can be used to design new regulatory sequences leading to desired expression levels[4–6]. State-of-the-art machine learning approaches have recently proven highly useful in this endeavor, expanding our knowledge of the DNA regulatory grammar underlying gene expression[7–10], helping us to design promoter and gene sequences[11,12] and accurately predict gene

expression across multiple model organisms[7,13]. The striking capacity of random DNA to evolve into functioning regulatory sequences by introducing only a small number of base pair mutations[14,15] suggests that the richness and plasticity of *cis*-regulatory grammar results in a vast functional regulatory sequence space, far larger than the one currently observed in natural systems[8]. By learning this regulatory sequence space using advanced deep learning approaches[11,16,17], we can in principle design systems that precisely traverse it to generate regulatory sequence variants with targeted expression levels.

Popular strategies to design synthetic regulatory DNA of varying expression levels include stacking multiple known functional sequence motifs[4,6,18,19] and applying random mutagenesis to a specific

[1]Department of Biology and Biological Engineering, Chalmers University of Technology, Kemivägen 10, SE41296 Gothenburg, Sweden. [2]Department of Biotechnology and Systems Biology, National Institute of Biology, Večna pot 111, SI1000 Ljubljana, Slovenia. [3]Department of Computer Science and Engineering, Chalmers University of Technology, Rännvägen 6, SE41296 Gothenburg, Sweden. [4]Biomatter Designs, Zirmunu st. 139A, LT09120 Vilnius, Lithuania. [5]BioInnovation Institute, Ole Maaloes Vej 3, DK2200 Copenhagen N, Denmark. [6]Institute of Biotechnology, Life Sciences Centre, Vilnius University, Sauletekio al. 7, LT10257 Vilnius, Lithuania. [7]Randall Centre for Cell & Molecular Biophysics, King's College London, New Hunt's House, Guy's Campus, SE1 1UL London, UK. [8]These authors contributed equally: Jan Zrimec, Xiaozhi Fu. ✉e-mail: jan.zrimec@nib.si; aleksej.zelezniak@chalmers.se

region, most commonly the promoter[8,20–23] though also UTRs[24–26] and terminators[27,28] have been targeted, typically in a form of short sequence segments of <100 bp. Using in silico screening approaches[7,8], which evaluate the fitness of candidate sequences by predicting their expression levels, more intricate solutions based on evolutionary computation[29] have been implemented, including genetic algorithms[15,24,29–31]. However, these algorithms still employ random mutagenesis in every round of sequence evolution, relying solely on the sequence-function mapping of the predictive models[29,32]. Rather than to generate valid sequences predicted to improve the target objective, they produce selection candidates via arbitrary sequence changes, many of which are not feasible regulatory DNA. This can potentially lead to highly untrustworthy predictions (predictor pathologies)[33–35] and local minima problems[32], exacerbating the difficulty of finding the small subset of sequences that satisfy the target objective in the enormous search space. Therefore, the search for functional sequence variants frequently requires experimental screening with multiple rounds of trial and error or experimentally testing enormous sequence batches[5,8]. The inherent inability in relating sequence to expression and the high resource intensiveness of the mutagenesis-based approaches are also the major factors constraining the explored DNA to only short segments of single regulatory regions and specific reporter genes[15,24]. This ultimately limits gene expression control, thus not fulfilling the key design objective.

Alternatively, the idea of novel solutions for regulatory DNA design, facilitated by deep neural networks, is to directly generate valid sequences by learning functional and biologically feasible sequence spaces[11,12,33,36]. This can resolve many mutagenesis-related problems and helps to optimize resources after the generative step, both in the case of experimental[11,37] or in silico screening[33,36], as sequence validity enables testing lower amounts of candidates and alleviates predictor pathologies[33–35]. However, despite not being restricted by sequence length, as no brute force testing of mutations spanning large sequence spaces is required, current generative approaches also focus on mere single regulatory regions[11] or shorter segments[17] and are rarely tested experimentally[33]. As evidenced by the strong agreement between protein and mRNA levels[38–41], mRNA transcription, a major determinant of protein abundance, is controlled by the interaction of cis-regulatory patterns across the whole regulatory structure of the gene. This comprises coding and regulatory regions that include promoters, untranslated regions (UTRs) and terminators, each encoding a significant amount of information related to mRNA levels[7,8,24,27]. Ultimately, to accurately control gene expression, the entire gene regulatory structure must be fine-tuned[3,7,42,43]. Therefore, based on the recent achievements in modeling DNA and protein spaces[11,12,44], we hypothesize that state-of-the-art generative deep neural networks are capable of learning the entire DNA regulatory landscape directly from natural genomic sequences and transcriptomic data. By leveraging information from the whole gene regulatory structure including the coding region[3,7], de novo regulatory DNA with highly accurate target expression levels can be generated, helping to overcome the limitations of existing approaches and enabling precise and gene-specific navigation of the regulatory sequence space in potentially any organism and tissue.

In the present study, we use deep learning frameworks to demonstrate that a generative modeling approach can successfully design de novo functional regulatory DNA in Saccharomyces cerevisiae. First, we train a deep generative adversarial network (GAN) only on natural genomic sequences spanning the whole gene regulatory structure and find that the generated regulatory sequences exhibit properties highly similar to those of natural regulatory DNA. Next, using an optimization procedure that couples the generative network with a highly accurate deep predictive model[7,17] (ExpressionGAN), we add coding sequence information to the generative approach and learn to precisely navigate the regulatory sequence-expression

landscape of a specific gene across almost 6 orders of magnitude of expression levels, accurately controlling the sampling of sequences with targeted expression levels. Similarly, we then train and optimize additional generators based on commonly used single regulatory region parts[15,24], demonstrating how the use of the whole gene regulatory structure can outperform single-region solutions by expanding the achievable dynamic range of expression levels. By sampling 20,000 generated regulatory sequences with high and low predicted expression levels from ExpressionGAN and measuring their sequence properties, including cis-regulatory grammar and core promoter features, we observe that the generated DNA carries known sequence determinants of gene expression control. Finally, we experimentally verify a selection of the generated sequences that retain a natural or even higher level of dissimilarity (>33%) to any currently known regulatory sequence. We find that experimentally measured mRNA expression levels recapitulate predicted ones across 3 orders of magnitude. In fact, 57% of the constructs designed to be highly expressed surpass the gene expression level of natural high-expression control sequences, demonstrating the effectiveness of the generative approach for designing functional regulatory DNA in practice.

## Results
### Implementing a generative strategy to design regulatory DNA
Based on the knowledge that the whole gene regulatory structure is involved in controlling gene expression[3], we previously demonstrated the combined DNA sequence across all regulatory regions (Fig. 1a: promoter, 5′ UTR, 3′ UTR and terminator) is highly predictive of gene expression[7]. We also observed that gene expression of individual genes varies across the majority of biological conditions within a mere 1-fold range for >80% of yeast protein-coding genes[7]. The dynamic range of gene expression (Fig. 1b: spanning nearly 5 orders of magnitude of median TPM values across the whole range of biological conditions) is thus predictable directly from the DNA sequence, irrespective of the biological conditions (Fig. 1c: $R^2_{test} = 0.8$, models tuned and tested on independent held-out datasets, see the "Methods" section). Apart from the properties of the coding region, the most relevant parts of DNA for these predictions were the respective sequences of the 4 regulatory regions totaling 1000 bp (as illustrated in Fig. 1a). Moreover, training and testing multiple deep neural networks using sequence data from different region combinations as input and median TPM values as the target (Fig. 1b) demonstrated that only the whole gene regulatory structure spans the key regulatory features important for predicting the full dynamic range of mRNA expression levels[7].

Driven by these findings, we aimed to improve upon current mutagenesis capabilities[15,30,45] with a more controlled approach for mapping sequence-function landscapes and designing synthetic regulatory DNA. We thus implemented a deep generative modeling strategy to mimic realistic regulatory sequence properties, by learning the genetic regulatory landscape comprising evolutionarily encoded expression rules and grammar directly from natural genomes. We trained a generative model (generator) using sequences spanning the whole gene regulatory structure (Fig. 1a) and a generative adversarial network (GAN) approach[46], where a discriminator network was used to train a generator, both comprising 6 convolutional layers (Fig. 1d, Supplementary Fig. 1, see the "Methods" section). As input data from which to learn the distribution of the gene regulatory sequence space, we used 4238 sequences spanning the promoter (Fig. 1a: 400 bp), UTRs (100 and 250 bp, respectively), and terminator (250 bp) from yeast. The data was split into training (90%) and testing datasets (10%), where the amount of training data was similar to that used in previous successful GAN implementations[16,47] and the testing data was used to represent natural sequences to evaluate model performance and control for overfitting. The performance of the generator was validated by quantitatively verifying that the sequence properties of the

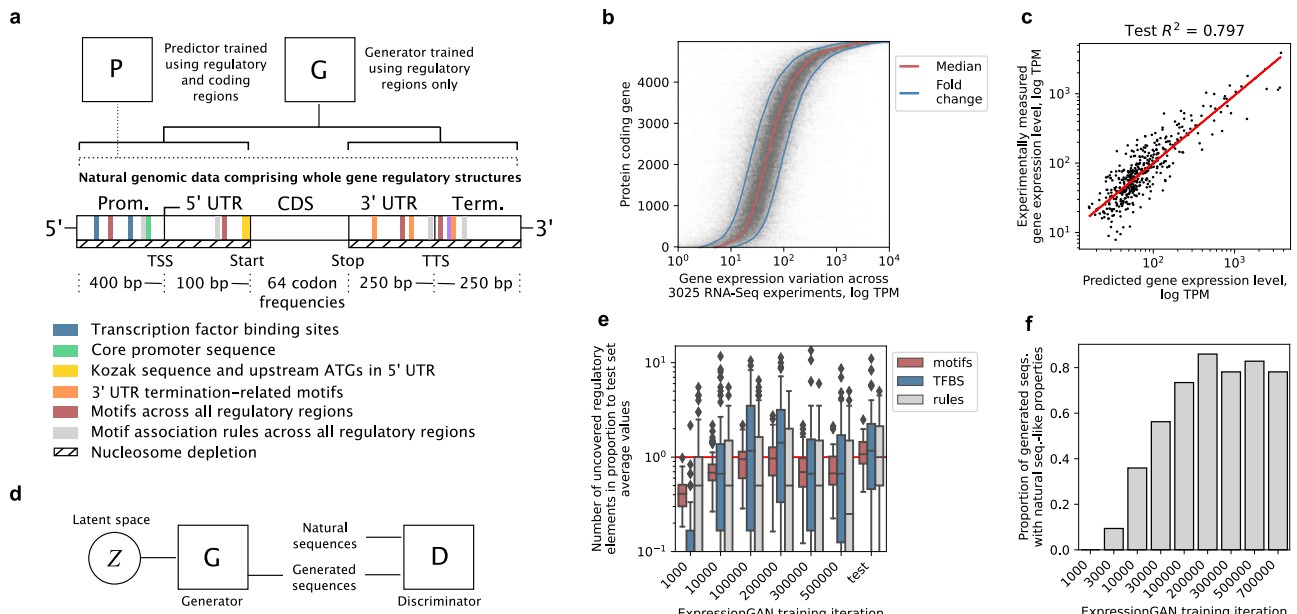

**Fig. 1 | Implementing a generative strategy to design regulatory DNA.**
**a** Schematic depiction of the *Saccharomyces cerevisiae* natural genomic sequencing dataset was used to train both the predictive (P)[7] and generative (G) models used in the study. The dataset spanned the whole gene regulatory structure of 1000 bp and included promoter, terminator, and untranslated regions (UTRs) as well as codon frequencies of coding regions. The different natural sequence properties related to DNA *cis*-regulatory grammar and further analyzed with the generator are indicted: transcription factor binding sites (TFBS, blue), core promoter elements (green), 5′ UTR elements (yellow), termination-related motifs (orange), deep learning-uncovered[7] motifs (red) and motifs association rules (gray), and nucleosome depletion (dashed lines) (see Supplementary Table 1 for a full list of tested seq. properties). **b** Median expression levels per gene (red line) derived from 3025 RNA-Seq experiments[59], with a 1-fold change marked in either direction (blue lines). **c** Performance of the deep predictive model of gene expression on the test dataset

($n = 424$), trained on natural genomic sequences spanning the whole gene regulatory structure. Red line denotes the least squares fit. **d** Overview of the generative adversarial network (GAN) approach, which iteratively trains a generative and discriminative deep neural network, the former learning to generate realistic sequences using random points in the latent space and the latter learning to discriminate between natural and generated sequences[46], resulting in a highly accurate generator. **e** Proportion of TFBS (blue), DNA motifs (red), and motif association rules (gray) in samples of generated sequences across generator training iterations ($n = 64$ each) relative to average amounts found in the natural test set. Red line denotes an equal amount. Boxes denote interquartile (IQR) ranges, centers mark medians and whiskers extend to 1.5 IQR from the quartiles. **f** Relative amount of generated sequences with properties similar to those of the natural test set (see Supplementary Table 1 and Supplementary Fig. 2). Source data are provided as a Source Data file.

generated variants reflected those of natural sequences present in the testing dataset (Fig. 1e, f, Supplementary Fig. 2, see the "Methods" section). The tested DNA sequence properties included: (i) sequence compositional validity, (ii) sequence similarity measures, and (iii) known *cis*-regulatory grammar (Fig. 1a, Supplementary Table 1). For instance, the average number of regulatory motifs including transcription factor binding sites (TFBS) and their combinations increased with the progression of training, peaking at 200,000 training iterations (Fig. 1e).

**Deep generative model designs natural-like regulatory DNA**
When studying the composition of the generated DNA after training (Fig. 1c: 200,000 training iterations of GAN), the majority of the synthetic sequences (86%) displayed properties similar to those of natural sequences (Figs. 1f and 2a, b). In each sequence, the model recapitulated not only the properties related to basic sequence composition, such as GC content (Fig. 2c: average GC content within 1.1% of values in natural sequences) and UTR sizes (Fig. 2d, e: average length of generated UTRs within 8 bp of natural ones) but also the known DNA regulatory grammar[3] (see Fig. 1a). This included (i) canonical motifs of transcription factor binding sites (TFBS) from the Jaspar database[48], identified (*q*-value < 0.05) using FIMO[49], and core promoter elements comprising the TATA box[50,51] in promoters (ii) Kozak sequences in 5′ UTRs[52,53], (iii) termination related motifs, including positioning, efficiency, and poly-AT motifs[6,54] in 3′ UTRs and terminators, (iv) previous deep learning-uncovered expression-related motifs and motif associations[7], as well as (v) positions predicted to be depleted of nucleosomes[55,56] (Fig. 2a, b, Supplementary Fig. 3, see the "Methods" section). We observed that on

average, the overall number of regulatory motifs and positions in the generated sequences (Fig. 2a, b), surpassed those found in natural test sequences by up to 28%, except for a ~4% decrease with core promoter elements (Supplementary Fig. 4). We also verified that the generated sequences retained a sequence diversity similar to that of natural sequences, with the average pairwise sequence identity of both the generated and test datasets to the training dataset equaling ~67% (Supplementary Fig. 2), showing that the nucleotide composition of generated variants was as variable and dissimilar to natural sequences as they were amongst themselves (Fig. 2f). This ensures that the model does not overfit the training dataset, simply reproducing it, and shows that the generator instead generalizes and designs de novo regulatory sequences with properties of natural regulatory DNA across the entire regulatory sequence landscape.

**Gene-specific navigation of the regulatory sequence-expression landscape**
Next, in order to exploit the generative model to produce regulatory DNA with target expression levels, we set up an optimization procedure[17,57] to guide sequence evolution (Fig. 3a). Briefly, we implemented a joint deep neural network architecture, termed ExpressionGAN, by coupling together the regulatory DNA sequence-generator and gene expression-predictor models within the thoroughly validated activation-maximization framework[17,32,58] (Fig. 3a). The principle behind this approach is to exploit the generator's ability to design regulatory sequences with realistic DNA properties by using the predictor to guide it and fine-tune generated sequences toward target gene expression levels. For the predictor, we used the

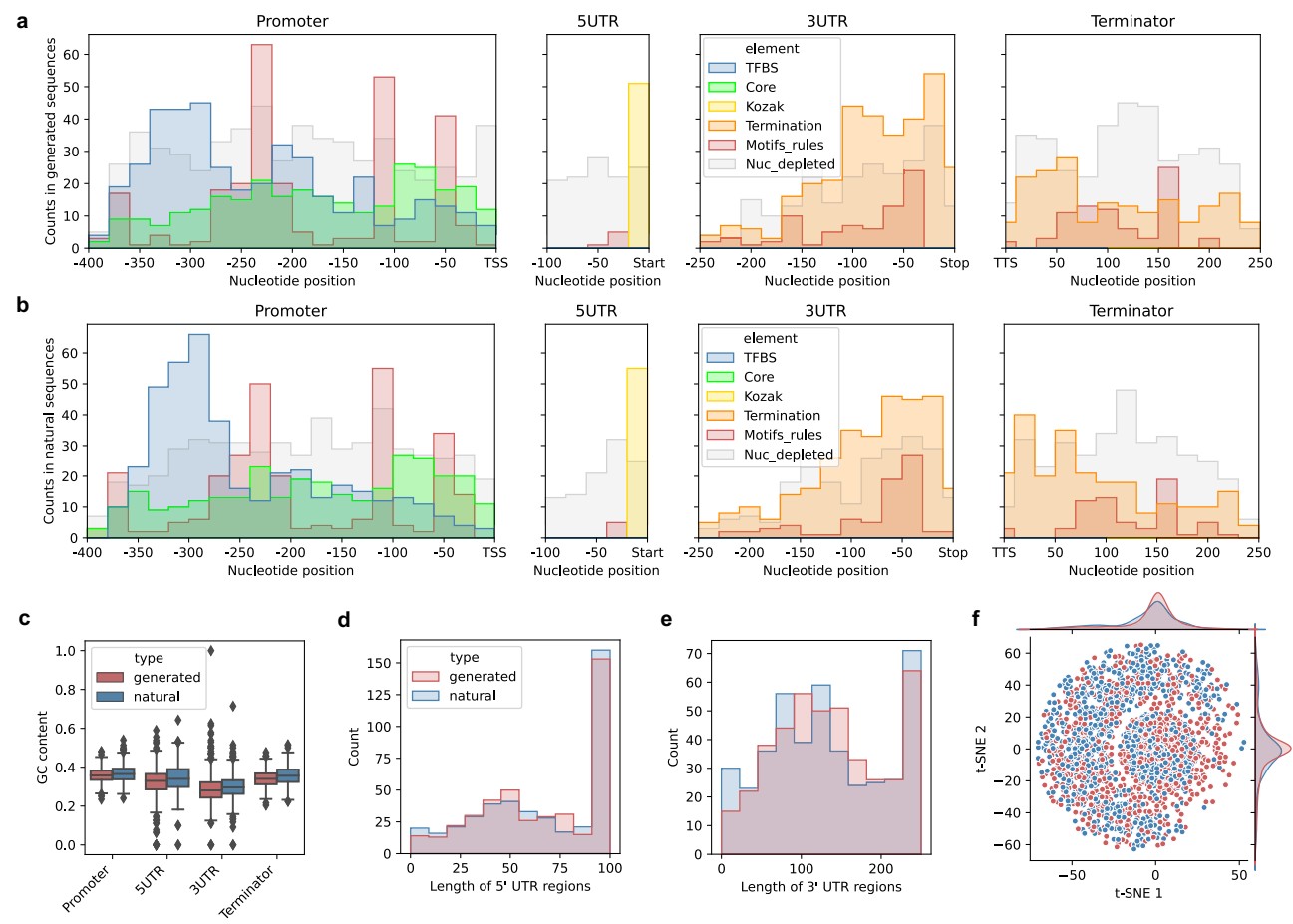

**Fig. 2 | Deep learning-generated sequences exhibit properties of natural regulatory DNA. a, b** Cumulative positional distribution of known DNA regulatory grammar elements (see Fig. 1a) across the regulatory regions of **a** generated synthetic and **b** natural sequences ($n = 425$ each). Shown are yeast TFBS[48] identified ($q$-value < 0.05) using FIMO[49] (blue) and TATA core promoter elements[50,51] (green) in promoters, Kozak sequences[52,53] in 5′ UTRs (yellow), termination related motifs (positioning, efficiency and poly-AT motifs)[6,54] in 3′ UTRs and terminators (orange), and deep learning-uncovered expression-related motifs and motif association rules[7] (red) as well as nucleosome depletion[55,56] (gray) across all regions. Note that the amount of Kozak sequences and nucleosome depleted positions are not shown to scale, with 4-fold and 200-fold dilutions, respectively, to improve visualization (see separate comparisons across elements in Supplementary Fig. 3). TSS denotes the transcription start site, Start/Stop the coding sequence start/stop positions and TTS the transcription termination site. **c** GC content in the equal-sized subsets of generated synthetic (red) and natural test sequences (blue) across the regulatory regions ($n = 425$ each). **d** Distribution of 5′UTR lengths in the synthetic (red) and (blue) natural sequences. Boxes denote interquartile (IQR) ranges, centers mark medians and whiskers extend to 1.5 IQR from the quartiles. **e** Distribution of 3′UTR lengths in the synthetic (red) and natural (blue) sequences. **f** T-distributed stochastic neighbor embedding (t-SNE) dimensionality reduction[60] over the sequence identity distance matrix among equal amounts of combined generated (red) and natural (blue) sequences ($n = 2000$ each). Source data are provided as a Source Data file.

experimentally validated highly-accurate model of yeast gene expression[7] ($R^2_{test} = 0.8$, Fig. 1c), trained on data derived from 3025 RNA-Seq experiments[59] with natural genomic sequences comprising whole gene regulatory structures used as explanatory variables to predict the target gene expression levels (Fig. 1a, b, model tuned and tested on independent held-out datasets). The trained generator and predictor neural networks were coupled in an optimization loop navigating the latent space of the generator to draw optimized sequence variants (Fig. 3b, see the "Methods" section). Since the predictor also evaluates variables describing the coding region (see Fig. 1a: 64 codon frequencies), this procedure in fact couples the generated regulatory sequences to a specific gene of interest.

Merging the results of both maximization and minimization of gene expression and using t-distributed stochastic neighbor embedding (t-SNE) dimensionality reduction[60] over the latent vectors confirmed that with this approach, desired expression levels are mapped to the identified latent subspace, resembling a continuous manifold, and covering regulatory sequence evolution in a range of almost 6 orders of magnitude of expression levels (Figs. 3c and 4a, b,

Supplementary Fig. 5). Thus, with optimization, the dynamic range of expression levels of the generated sequences increased over 3-fold compared to those obtained by randomly sampling the unoptimized generator (in equally sized samples), surpassing the whole natural range of expression levels (Fig. 1b: 4 orders of magnitude of median TPM across conditions), entirely for a specific gene of interest, by ~40% (Fig. 3c: GFP coding sequence shown). As before, analysis of sequence identity verified that the sequences produced by the generator optimization were not similar to any natural ones and retained the natural median sequence diversity of 67% (Supplementary Fig. 6). This suggested that points in the generator's latent space with desired expression levels can be sampled, which generalize beyond the naturally available gene expression levels, to generate unique sequences with feasible natural-like regulatory DNA properties.

## Enhanced control of expression using whole gene regulatory structure

In order to compare our approach with existing solutions and determine whether using the whole gene regulatory structure outperforms

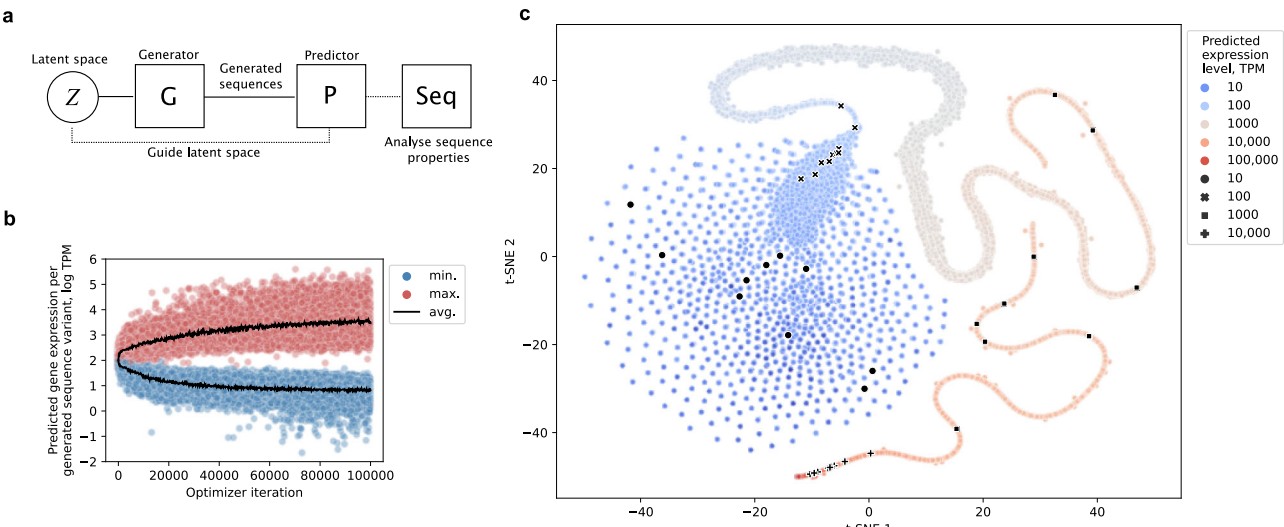

**Fig. 3 | Predictor-guided generator optimization enables gene-specific navigation of the regulatory sequence-expression landscape. a** Schematic depiction of the procedure to optimize the generator using a trained predictor[7], which introduces codon frequency information into the generative approach and explores the input latent space of the generator to produce sequence variants across the whole range of gene expression, providing precise navigation of the gene regulatory sequence-expression landscape. **b** Predicted expression levels of generated sequence variants across optimization iterations set to either maximize (red) or minimize (blue) expression levels (*n* = 64,000). Black lines denote average expression levels and TPM transcripts per million. **c** *T*-distributed stochastic neighbor embedding (t-SNE)[60] mapping of the input latent subspaces that produce unique sequence variants spanning ~6 orders of magnitude of gene expression (black and colored dots: progression of low to high expression levels is marked with progression from blue to red, respectively), uncovered using the predictor-guided generator optimization. Black dots represent selections of 10 sequence variants per each of the 4 expression groups covering a 4 order-of-magnitude range of predicted expression levels from TPM ~10 to ~10,000. Source data are provided as a Source Data file.

solutions based on single regulatory regions, we next trained and optimized six additional generative models using the same procedures as for ExpressionGAN detailed above (see the "Methods" section). In addition to using whole single regions in the respective ranges defined above (see Fig. 1a), specifically the promoter (400 bp), UTRs (100 and 250 bp, respectively) and terminator (250 bp), we also used two shorter parts of the promoter featured in recent studies[15,31]. These included an 80 bp proximal promoter region located between −170 and −90 bp upstream of the transcription start site (TSS)[8,15] and the core promoter region located −170 bp upstream up to the TSS[31,61]. We compared the dynamic ranges between either median or extreme predicted expression levels in generated sequence samples after 100,000 optimizer iterations of maximization and minimization (Fig. 4a, b). Interestingly, we observed that out of the single region generators, the terminator-based generator showed the highest expression range of ~3 orders of magnitude, whereas the 5′ UTR and 80 bp proximal promoter-based generators resulted in the lowest ranges of ~1 and ~2 orders of magnitude, respectively (Fig. 4b: values approximate both median and maximum ranges). Importantly, the dynamic range of ExpressionGAN (generator based on the whole gene regulatory) was from 29% to 277% larger in the case of median expression values with best performing (terminator) and worst performing (5′ UTR) generator variants, respectively (Fig. 4b), reflecting a 6 to 358-fold increase between median expression levels of maximization- and minimization-based sequence samples (Fig. 4a). This was further supported by analyzing the relevance of different regulatory region combinations (Supplementary Fig. 7), showing that the regions jointly contribute to gene expression control[3].

Utilizing available published proximal promoter[15] and 5′ UTR sequence designs[24], we explored the potential of using the whole gene regulatory structure to unlock a wider range of gene expression than is achievable by using the shorter sequence designs. When comparing published expression measurements and our predictions for a set of 80 bp proximal promoters, recently designed using a genetic algorithm under a strong-selection weak-mutation regime yielding both

maximization and minimization of expression levels[15], we observed a good correlation (Spearman's ρ = 0.51, *p*-value < 1e−16, Fig. 4c). As ExpressionGAN was found to achieve a 2-fold higher median expression range than the proximal promoter-solution (Fig. 4a, b), we randomly sampled 128 of these proximal promoter sequence designs (from both maximized and minimized expression groups)[15] and expanded them with all 4238 available native gene regulatory structures, yielding 542,464 sequence constructs that were used to analyze any additional dynamic potential with the already optimized short sequences. Indeed, we observed that a dynamic range spanning an order of magnitude of predicted expression levels was achievable (Fig. 4d: between 10th and 90th percentiles of expression levels). We next performed a similar experiment with a set of 5′ UTRs also designed using a genetic algorithm[24], selecting sequences at their respective optimal number of evolutionary rounds yielding the highest predicted protein expression levels as estimated from cell growth. Similarly, as above, we observed a good correlation (Spearman's ρ = 0.55, *p*-value < 1e−16) between the published growth rates and our expression predictions (Fig. 4e). Moreover, by building whole gene regulatory structures around a random subset of 128 of the 5′ UTR sequence designs, yielding 542,464 sequence constructs, we again observed that a range of over an order of magnitude of predicted expression levels could still potentially be unlocked (Fig. 4f: between 10th and 90th percentiles of expression levels). This suggests that, generally, short single-region sequence designs might not be capable of controlling gene expression across its full dynamic range. Thus, despite the sequences being optimized in their restricted sense[15,24], gene expression cannot be driven to its actual extremes without proper optimization of the gene regulatory structure with all adjacent regulatory regions.

## Generated regulatory DNA carries sequence determinants of gene expression control

Next, we asked which sequence features of the generated synthetic sequences can be inferred to control gene expression and how the *cis-*

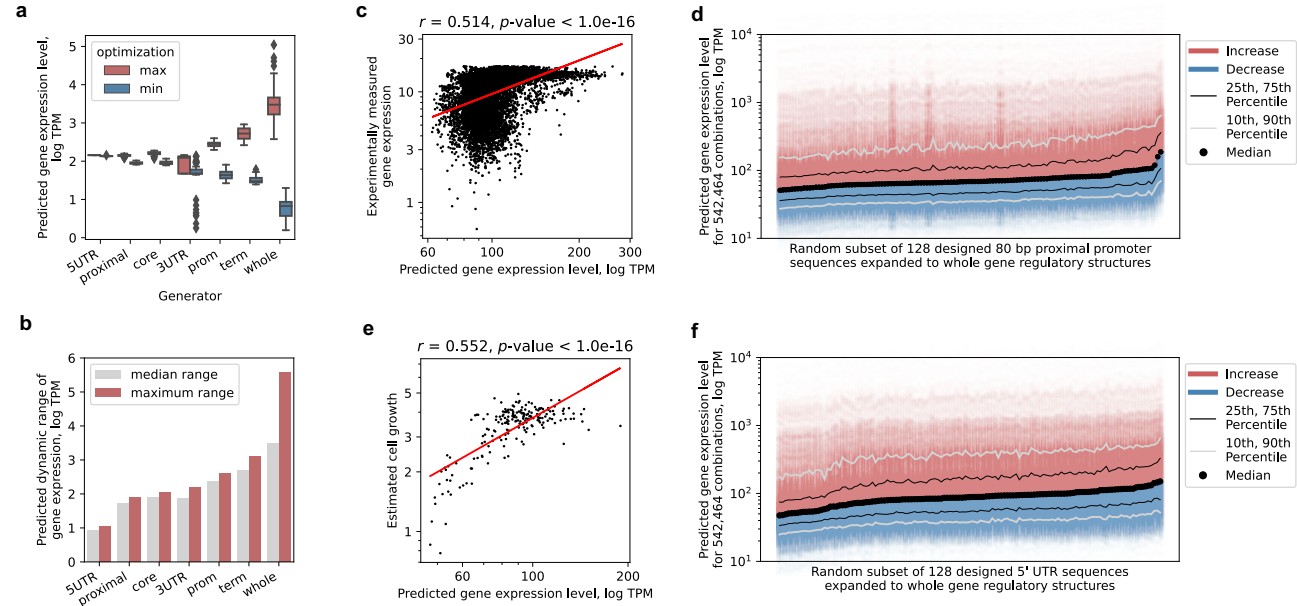

**Fig. 4 | Whole gene regulatory structure unlocks a wider range of expression control than single regulatory regions. a** Predicted gene expression levels with optimized generators of different single regulatory region parts or sequences spanning the whole gene regulatory structure ($n = 64$ per specific generator optimization target sample, maximization marked red, minimization blue). Boxes denote interquartile (IQR) ranges, centers mark medians and whiskers extend to 1.5 IQR from the quartiles. **b** Dynamic ranges between median (gray) and extreme values (red) in the optimized sequence samples from **a**. **c** Correlation analysis between published experimentally measured gene expression levels (defined medium) of 80 bp proximal promoter sequences ($-170$ to $-90$ relative to TSS)[15] and our predictions ($n = 10,282$). Red line denotes the least-squares fit. The $T$-test was used. **d** Increases (red) and decreases (blue) of predicted gene expression levels

with a random subset of the 80 bp proximal promoter designs[15] when expanded and combined with all 4238 native gene regulatory structures to create 1000 bp constructs ($n = 542,464$). Black dots denote median levels, black lines the inter-quartile range and gray lines the 10th and 90th percentiles, respectively. **e** Correlation analysis between published estimated cell growth of 5′ UTR designs (at the optimal level of evolutionary rounds)[24] and our predicted gene expression levels ($n = 200$). Red line denotes the least-squares fit. The $T$-test was used. **f** Increases (red) and decreases (blue) of gene expression levels with a random subset of the 5′ UTR designs[24] when expanded and combined with all 4238 native gene regulatory structures to create 1000 bp constructs ($n = 542,464$). Black dots denote median levels, black lines the interquartile range and gray lines the 10th and 90th percentiles, respectively. Source data are provided as a Source Data file.

regulatory grammar might interact to define expression levels. For this, high and low expression levels were contrasted by sampling ExpressionGAN-generated sequences after 50,000 optimizer iterations (Fig. 3b) and segregating them based on the strength of predicted expression levels into high (TPM > 100) and low expression bins (TPM < 100) of 10,000 samples each (Fig. 5a), also verifying that they were valid and different from any natural sequence. We observed a significant difference (Wilcoxon rank-sum test $p$-value < 1e−16) in GC content between the high and low expression sequences, with promoters and terminators displaying a 7% higher and UTRs a 12% lower GC content in the high expression sequences compared to low expression ones (Fig. 5b, c). In accordance with the knowledge that UTR length is related to mRNA stability[9], a 28% (Wilcoxon rank-sum test $p$-value < 1e−16) increase in 5′ UTR size was measured between the high and low expression sequences (Supplementary Fig. 8). Furthermore, as expected based on the previous findings[7,18], we observed on average a 67% higher amount (Wilcoxon rank-sum test $p$-value < 1e−16) of identified promoter TFBSs (FIMO[49] $q$-value < 0.05) in the high expression sequences compared to low expression ones (Supplementary Fig. 9), with the number of TFBS significantly correlated (Spearman's $\rho = 0.36$, $p$-value < 1e−16, respectively) with the predicted gene expression levels (Fig. 5d, e). The type of TFBS was however not found to be related with and thus indicative of the expression levels, though a weak correlation (Spearman's $\rho = 0.13$, $p$-value < 1e−16) was observed between average expression levels of natural genes where a TFBS was found and the predicted expression level of the generated sequence carrying it (Supplementary Fig. 10). Previous studies have shown that, apart from transcriptional regulation, Kozak sequence[62] composition also affects overall mRNA levels by regulating mRNA

degradation rates[25,63,64], with adenine enrichment resulting in higher levels of gene expression[52]. Accordingly, we detected a moderate correlation (Spearman's $\rho$ between 0.44 and 0.51, $p$-value < 1e−16) between the number of adenines in the 5–15 bp region upstream of the start codon, respectively, and predicted gene expression levels (Supplementary Fig. 11). On average, high expression sequences carried 87% more (Wilcoxon rank-sum test $p$-value < 1e−16) adenines than low expression sequences (Fig. 5f). Similarly, in the 3′ UTR and terminator regions, both the presence and number of poly-A/T, positioning (consensus 5′-AAWAAA-3′)[54,65] and efficiency motifs (5′-TATDTA-3′)[27] were correlated (Spearman's $\rho$ between 0.06 and 0.51, $p$-value < 1e−16, respectively) with the predicted expression levels (Supplementary Fig. 12), with a 9%, 6%, and 45% higher amount of motif-carrying sequences, respectively, found in the high expression sequences compared to low expression ones[26] (Fig. 5g). The numbers of known expression-related motifs and motif association rules[7] across all regulatory regions were also found to be 26% and 125% higher (Wilcoxon rank-sum test $p$-value < 1e−16), respectively, in the high expression sequences compared to low expression ones[7,18] (Fig. 5e), and thus were significantly correlated (Spearman's $\rho$ was 0.36 and 0.26, $p$-value < 1e−16, respectively) with the predicted expression levels (Supplementary Fig. 13). This suggested that, for controlling gene expression levels, the optimized ExpressionGAN attributes at least partial relevance to each known DNA sequence feature, with full expression control likely achieved by combining multiple relevant sequence properties[8,9,28].

Since the core promoter is arguably the most crucial DNA element in transcriptional regulation[7,61], we next verified if our generative model also recapitulates the main properties of this region (Fig. 5c). In

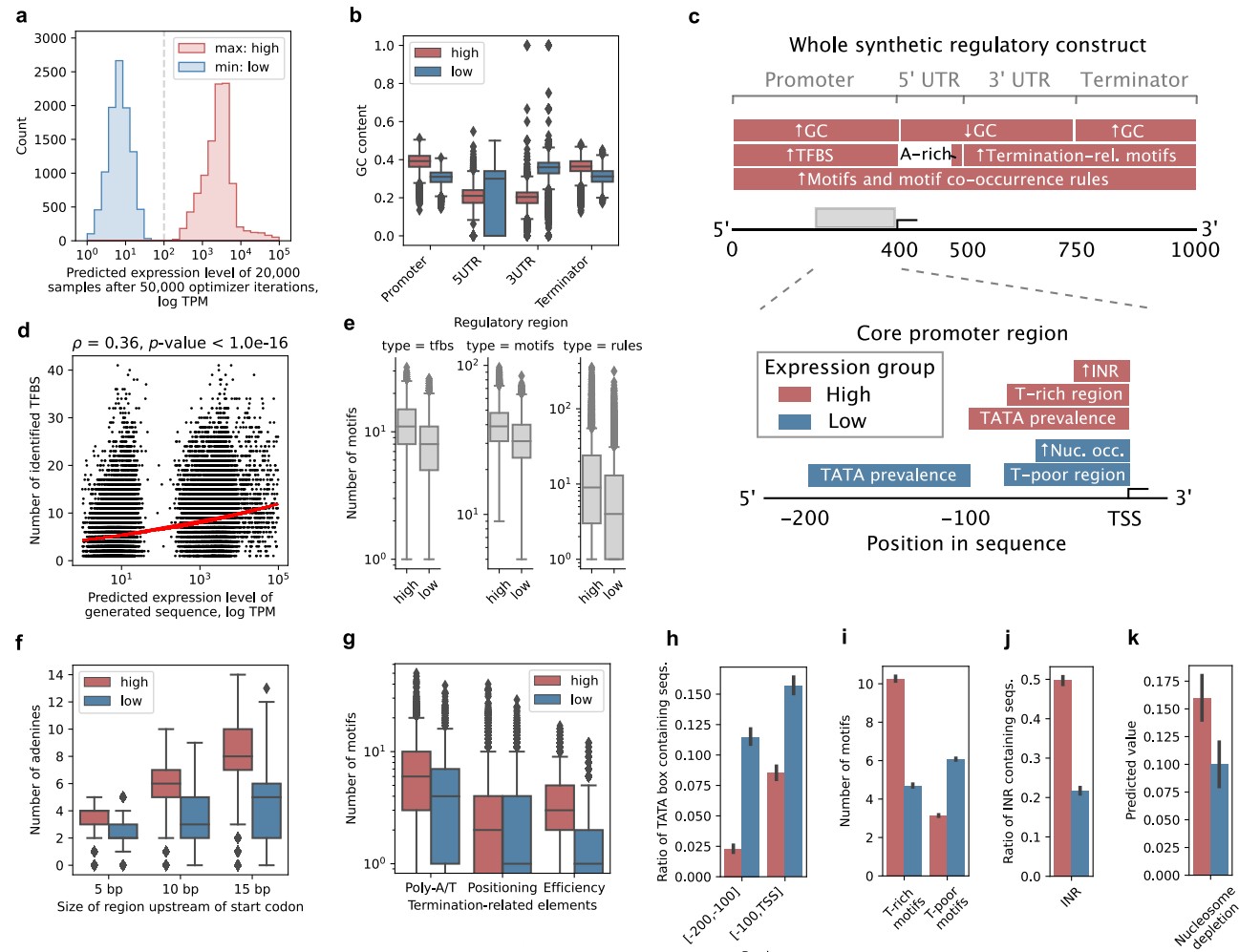

**Fig. 5 | ExpressionGAN-generated regulatory DNA carries sequence determinants of gene expression control. a** Distribution of predicted expression levels of generated sequence samples from low (blue) and high (red) expression bins ($n = 10,000$ each) after 50,000 iterations of ExpressionGAN optimization (see Fig. 3). TPM denotes transcripts per million. **b–k** Panels display sequence properties of the generated sequences from the low (blue) and high (red) expression bins. **b** GC content across the regulatory regions of the generated sequences ($n = 10,000$ each). **c** Overview of findings across the whole generated synthetic regulatory constructs as well as the core promoter regions. Nuc. occ. denotes higher nucleosome occupancy. **d** Correlation analysis between the amount of identified TFBS[48] (FIMO[49] $q$-value < 0.05) in promoters and predicted expression levels of the generated sequences ($n = 20,000$). Red line denotes the least squares fit. The $T$-test was used. **e** Amount of yeast transcription factor binding sites (TFBS) and deep learning-uncovered expression-related motifs and motif association rules[7]

($n = 10,000$ each). **f** Amount of adenines conserved in 5, 10, and 15 bp 5′ UTRs upstream of the start codon[52] ($n = 10,000$ each). **g** Number of termination-related elements, including Poly-A/T, positioning and efficiency motifs[27,54,65] ($n = 10,000$ each). **h** Proportion of sequences carrying a conserved TATA box[50] in the distal and proximal parts of the core promoter region[51] ($n = 10,000$ each) (Fig. 5c). **i** Amount of T-rich and T-poor motifs in the region up to 75 bp upstream of the TSS[51,68] ($n = 10,000$ each). **j** Proportion of mammalian-type INR motifs in the region up to 30 bp upstream of the TSS[69] ($n = 10,000$ each). **k** Proportion of predicted nucleosome depletion[55,56] in the region up to 50 bp upstream of the TSS[70,71] ($n = 10,000$ each). For box plots in **b**, **e**–**g**, boxes denote interquartile (IQR) ranges, centers mark medians and whiskers extend to 1.5 IQR from the quartiles. For bar plots in **h**–**k**, error bars represent 95% confidence intervals. Source data are provided as a Source Data file.

yeast, ~20% of genes are known to contain a strong TATA box (consensus 5′-TATAWAWR-3′)[50], whereas the large majority of the remaining genes carry a TATA-like element[66,67] that differs from the TATA consensus by up to 2 bases and can still recruit the TATA-binding protein (TBP), though with lesser affinity[67]. In accordance with this, we observed that over 18% of the generated sequences contained at least a single TATA consensus motif and 99% of them contained a TATA-like sequence in their core promoters. Whereas the high expression sequences carried an almost 4-fold and significantly (Wilcoxon rank-sum test $p$-value = 1.9e−14) higher number of TATA motifs in the [−100, TSS] region compared to the [−200, −100] region, low expression sequences carried a 5-fold (Wilcoxon rank-sum test $p$-value < 1e−16) higher number of TATA motifs in the [−200, −100] region compared to the high expression sequences (Fig. 5h, Supplementary Fig. 14).

Another prominent core promoter feature is the general richness of thymine in the −75 to TSS region as well as the presence of T-rich and T-poor 4-mer motifs (see the "Methods" section)[51,68]. Indeed, we observed a 39% higher (Wilcoxon rank-sum test $p$-value < 1e−16) T-richness in the high expression sequences (Supplementary Fig. 14), carrying over 2-fold more (Wilcoxon rank-sum test $p$-value < 1e−16) T-rich elements, compared to low expression sequences that contained 2-fold more (Wilcoxon rank-sum test $p$-value < 1e−16) T-poor elements (Fig. 5i, c). Moreover, in accordance with previous findings[69], mammalian-like INR sequences (consensus 5′−'YYANWYY'−3′) that are known to cluster around the TSS region were found in 36% of the generated core promoters. There were over 2-fold more (Wilcoxon rank-sum test $p$-value < 1e−16) high expression sequences with at least one INR motif than low expression sequences (Fig. 5j; region −30 to

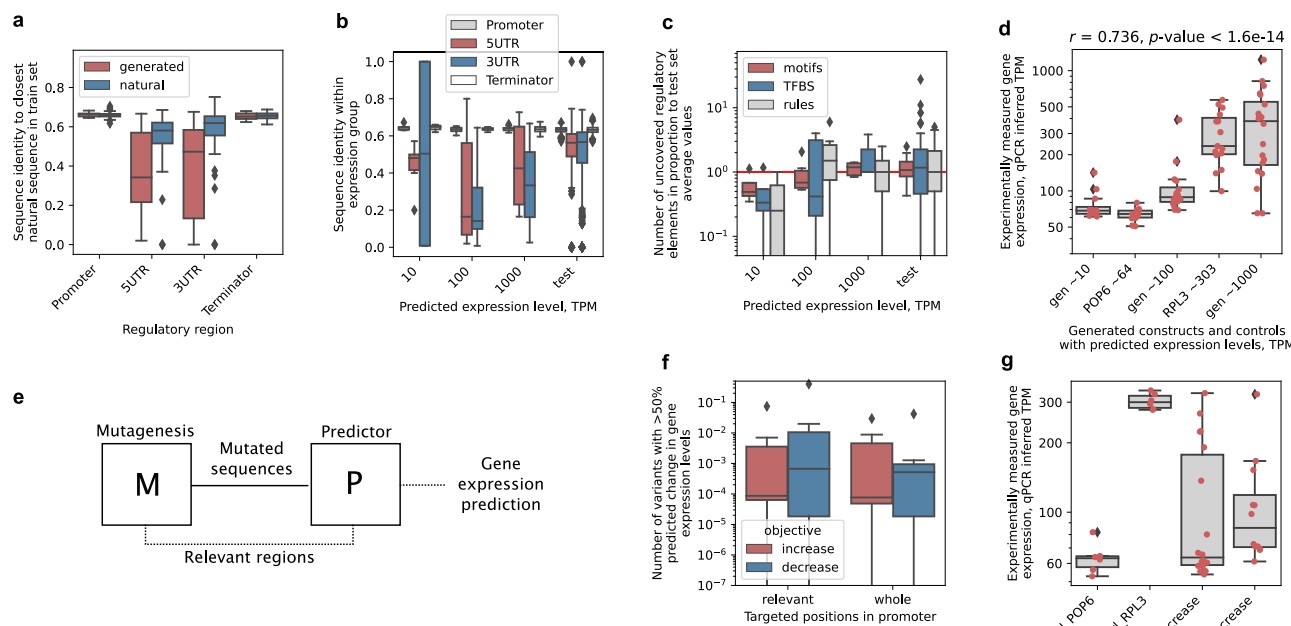

**Fig. 6 | Gene expression control using generated regulatory DNA is validated in vivo. a** Sequence homology of the experimentally validated variants produced by generator optimization (red) (see Supplementary Table 2) and natural test sequences (blue) to the respective closest representative sequences in the training dataset across the 4 regions of the gene regulatory structure ($n = 81$ each).
**b** Sequence homology within the experimentally validated expression groups spanning 3 orders of magnitude of predicted expression levels (TPM of ~10, ~100 and ~1000, Supplementary Table 2; $n = 12, 18, 21$, respectively) as well as the natural test set ($n = 192$), across the 4 regions of the gene regulatory structure: promoter (gray), 5′ UTR (red), 3′ UTR (blue), terminator (white). **c** Proportion of TFBS (blue), DNA motifs (red) and motif association rules (gray)[7] in the experimentally validated generated sequence variants ($n = 12, 18, 21$, respectively) relative to average amounts found in the natural test set ($n = 192$). Red line denotes equal amount to natural test set. **d** Quantitative PCR (qPCR) measurements of mRNA levels with groups of generated sequence variants across 3 orders of magnitude of predicted expression levels (TPM of ~10, ~100 and ~1000, Supplementary Table 2; $n = 12, 18, 21$, respectively). Natural regulatory regions of the POP6 and RPL3 genes were used

as low and high controls with a predicted TPM of 64 and 303, respectively ($n = 15$ each). Spearman correlation coefficient and *T*-test results shown. **e** Schematic depiction of the mutagenesis strategy that included in silico screening, where a random mutagenesis procedure (M) was coupled with a predictor (P) of yeast gene expression[7], which was also used to inform the mutational procedure on which positions were the most relevant to mutate. **f** Amount of mutated sequence variants that achieved an over 50% increase (red) or decrease (blue) in predicted gene expression levels by mutating 10% (40 bp) of whole promoter regions (400 bp) or only the most relevant promoter positions ($n = 14$ each). **g** Quantitative PCR (qPCR) measurements of mRNA levels with 10 mutated RPL3 sequence variants predicted to achieve ~2-fold increases ($n = 18$) or decreases ($n = 12$) in expression levels from the native regulatory sequence (see Supplementary Table 2). Native regulatory regions of the RPL3 and POP6 genes were used as high and low expression controls, respectively (predicted TPM of 303 and 64, respectively; $n = 6$ each). For box plots in **a**–**d**, **f**, **g**, boxes denote interquartile (IQR) ranges, centers mark medians and whiskers extend to 1.5 IQR from the quartiles. Red dots in **d**, **g** show separate measurements. Source data are provided as a Source Data file.

TSS) and a 44% more frequent (Wilcoxon rank-sum test *p*-value < 1e−16) occurrence of INR in promoters lacking a strong TATA box[69] (Supplementary Fig. 14). Finally, based on nucleosome occupancy predictions[55,56] and as expected based on the fact that nucleosome depletion aids promoter accessibility and transcription activation[70,71], positions in the [−50, TSS] region displayed on average 60% higher (Wilcoxon rank-sum test *p*-value < 1e−16) nucleosome depletion in the high expression sequences compared to the low expression sequences (Fig. 5k). This demonstrated that ExpressionGAN recapitulates the main functional properties of the core promoter (Fig. 5c) with the combined results suggesting that it can recreate both low and high expression-related properties across the whole gene regulatory structure in accordance with existing knowledge.

**In vivo gene expression control using generated regulatory DNA**
In order to test the validity and efficacy of the procedure and to verify that the generated sequence variants are active in vivo and correspond to the predicted gene expression levels, we selected a group of generated sequences across a 4 order-of-magnitude range of predicted expression levels (Fig. 3c: TPM of ~10, ~100, ~1000, and ~10,000). For this, since the prior results showed that not all sequence properties were strongly divergent among the different expression levels (e.g. similar UTR lengths among high and low expression sequences, see Supplementary Fig. 8), we used a selection procedure that retained

sequences with properties similar to those of natural sequences in each respective expression range to remove outliers (see the "Methods" section). The exception was that sequences that with the lowest sequence similarity to natural ones were preferred (Fig. 6a, Supplementary Fig. 15) and, importantly, sequence diversity was maximized in each expression range so that no two tested sequences were alike (Fig. 6b, Supplementary Fig. 15), in order to test a wide range of unique sequence variants and not merely multiple versions of a common variant. Moreover, to ensure proper core promoter function by facilitating TATA-binding protein (TBP) interaction[67], we retained only sequences carrying TATA or TATA-like motifs with up to 2 mutations from the TATA sequence consensus. The resulting generated sequence selections displayed properties similar to those of natural sequences (Supplementary Fig. 16, Fig. 1a, Supplementary Table 1), indicating that they are potentially functional, with the overall amounts of *cis*-regulatory motifs observed to steadily increase in proportion to the predicted gene expression levels (Fig. 6c). Unfortunately, as a result of the restrictions imposed by sequence synthesis technology, limiting the possibility to synthesize very lowly or highly-expressed sequences (<10 and >1000 TPM), we succeeded in testing 17 regulatory sequence-GFP constructs across the 3 order-of-magnitude range of expression levels from ~10 to ~1000 TPM (Figs. 3c and 6d, see the "Methods" section). Additionally, regulatory sequences of the POP6 (predicted TPM of 64) and RPL3 (predicted TPM of 303) genes[7] were used as low and high

controls, respectively. Although mRNA levels were measured, the GFP gene was used due to its low effect on cell growth (Supplementary Fig. 17).

We observed that experimental measurements of the mRNA levels produced by each construct achieved a strong correlation with the predicted levels (Spearman's ρ = 0.74, p-value < 1.6e−14, Fig. 6d, Supplementary Fig. 18). The measured levels were also significantly correlated (Spearman's ρ between 0.37 and 0.75, p-value < 1e−2) with specific regulatory sequence properties including the number of cis-regulatory motifs[7], 5′ UTR length[9,72], nucleosome depletion[55,56] and presence of an A-rich Kozak sequence[52] (Supplementary Fig. 19), supporting our findings that the generated DNA carries sequence determinants of gene expression control (Fig. 5). Despite this, on average, the measured expression levels reflected the predicted expression range only in the group of constructs with the predicted TPM of ~100 (Fig. 6d: avg. measured TPM of 114), whereas a 7.7-fold and 2.5-fold difference between predictions and measurements was observed with the lower (predicted TPM ~10, avg. measured TPM 77) and higher groups (predicted TPM ~1000, avg. measured TPM 397), respectively. Nevertheless, although we were unable to generate sequences with expression lower than the POP6 control, within the highest expression group, 4 out of 7 regulatory constructs (57%) displayed average expression levels that surpassed those of the natural highly-expressed RPL3 control by up to 2.7-fold (Fig. 6d, Supplementary Table 2). This demonstrates that our generative procedure enables the design of de novo gene regulatory DNA that exceeds native highly expressed genes by learning the natural regulatory DNA variation directly from genomic data, without relying on conventional experimental screening.

Finally, a primary advantage of ExpressionGAN is that it constrains the exploration of regulatory sequence space using natural regulatory principles incorporated in the generator, thus providing the predictor with feasible and biologically-consistent sequence candidates (Fig. 3a). On the other hand, mutational approaches, such as random mutagenesis[8,14,23] and genetic algorithms[15,30], rely solely on a predictive model to test the fitness of arbitrary and potentially infeasible mutations, which can lead to highly untrustworthy results[33–35]. To test the performance of such a rudimentary sequence design approach, we used conventional random mutagenesis coupled with in silico screening by the gene expression-predictor[7] (Figs. 1c and 6e, see the "Methods" section). We designed sequences spanning the whole gene regulatory structure as with the generative approach above, but since the whole accessible dynamic range was not being tested and thus full-scale mutagenesis was unnecessary, the UTR and terminator regions were kept intact and we focused on mutating either the whole promoter region (Fig. 1a: 400 bp[7]) or only the most relevant parts of it (Supplementary Fig. 20: 77 bp on average per sequence, Fig. 6e). To sustain predictor functionality and avoid pathologies, as sequences far outside the biologically allowed sequence space could push it into unpredictable behavior, we controlled the amount of mutated sequence between 1% and 10%. By creating and assessing 16.8 million sequence variants at different parameters using regulatory sequences of 7 natural genes as starting points (Supplementary Fig. 21), we obtained a distribution of sequence variants with different amounts of mutations (Supplementary Fig. 21: 1%, 2%, 5%, and 10%) and predicted expression levels, enabling the further selection and testing of variants that achieved desired expression changes.

When aiming to achieve an over 50% increase or decrease in mRNA expression levels, we found that on average, at most 0.3% of the sequence variants were predicted to achieve the desired effect when mutating 10% (40 bp) of whole promoter regions (Fig. 6f). This increased to 0.4% when mutating the most relevant promoter regions (Fig. 6f), whilst greatly decreasing with lower percentages of mutated sequence size (Supplementary Table 3). Analogously as for the validation of ExpressionGAN, we selected and experimentally tested 10 of the mutated regulatory sequence variants of the RPL3 gene (among the highest expressed genes in yeast) with the largest predicted (~2-fold) increase or decrease from the native levels, including both whole and only relevant-region mutational strategies and different percentages of mutated sequence size (5% and 10%, Supplementary Table 4, see the "Methods" section). Of the tested variants, 40% corresponded with predictions, and none of these were variants designed to increase expression levels but only to decrease them (Fig. 6g, Supplementary Fig. 22). This indicates that sequences designed by the random mutagenesis approach are unlikely to function as predicted, especially when trying to increase expression levels, necessitating multiple rounds of selection and experimental testing despite the use of in silico screening. In contrast, operating fully within the biologically feasible sequence space, ExpressionGAN can generate highly divergent constructs (Fig. 6a, b: avg. pairwise seq. identity of ~67% or lower) that achieve target expression levels, without requiring subsequent experimental screening.

## Discussion

In the present study, we explored whether de novo functional regulatory DNA, spanning the whole gene regulatory structure and producing desired gene expression levels, can be generated just from the knowledge of natural regulatory sequences. There are over $10^{60}$ ways to construct a mere 100 bp promoter sequence, covering more DNA variation than exists in all living species on our planet. Experimentally exploring even a tiny fraction of such an enormous sequence space is challenging and often infeasible due to the vast species diversity and complexity of eukaryotic gene regulation. Here, we thus used state-of-the-art deep learning models[7,17,46,57,73] to learn and map the functional DNA regulatory sequence space to gene expression levels directly from natural genomic data in Saccharomyces cerevisiae, enabling the design of expression systems in a controlled manner.

This was made possible by incorporating multiple recent advances that enabled us to develop our supervised deep generative modeling approach: (i) sequences spanning the whole gene regulatory structure[3] within natural genomic datasets[7] (Fig. 1a: 1000 bp across all cis-regulatory regions), (ii) highly accurate predictive models of gene expression levels that can explain over 82% of expression variation from regulatory sequence alone[7] (Fig. 1b), (iii) deep generative modeling procedures that are capable of learning and expanding functional coding[12,37] and regulatory[11,17,33] sequence spaces from natural genomic data (Fig. 1d), and (iv) optimization procedures that are thoroughly validated[17,57,73] and allowed us to include coding region information in sequence design, enabling gene-specific fine-tuning of generators across the whole range of expression levels (Fig. 3a, c). With the latter, due to the possibility to connect deep neural networks in end-to-end differentiable architectures, the existing capability of deep generative models to learn the DNA regulatory sequence space from natural genomic data[11] was expanded using optimization guided by predictive models (Fig. 3a). This enabled us to gain control over the generator's mapping of the regulatory sequence space to the respective expression levels and navigate the functional regulatory sequence-expression landscape, to produce generated sequences with desired expression levels in a range of almost 6 orders of magnitude of TPM (according to computational predictions, Figs. 3b, c and 4a, b). We can thus design unique regulatory sequence variants (Fig. 6a, b) that are nevertheless functional and contain natural-like properties and cis-regulatory grammar (Figs. 2 and 5), even surpassing the expression level of natural highly-expressed genes (Fig. 6d). Moreover, since our DNA-generator has learned the generalized functional regulatory sequence space, it can generate a practically infinite supply of unique sequence samples for any gene. It traverses only the most relevant sequence subspace instead of randomly sampling candidates from all $4^{1000}$ possible sequence variants, which would otherwise be needed to explore the 1000 bp of regulatory DNA. Therefore, by advancing generative models to span the whole gene regulatory structure and by mapping

sequence generation directly to the entire dynamic range of expression levels, we present a validated solution for gene expression control in eukaryotic species.

We tested different functional aspects of ExpressionGAN and compared it with existing solutions experimentally or computationally: (i) experimentally validating ExpressionGAN-generated sequences (Fig. 6), (ii) comparing the use of the whole gene regulatory structure to single regulatory regions and shorter promoter parts, commonly used with mutagenesis[15,24,31] (Fig. 4), (iii) comparing the properties of generated sequences to natural ones (Fig. 2) and testing whether they contain known DNA regulatory grammar and properties that drive gene expression (Fig. 5), and (iv) contrasting the generative approach with a standard mutational one that does not inherently model the allowed sequence-function landscape (Fig. 6). The experimental analysis was designed specifically to test the most divergent possible sequence variants, with an average sequence identity well below 70% both to natural sequences and amongst themselves (Fig. 6a, b, Supplementary Fig. 15). This validated the approach in vivo across 3 orders of magnitude of expression levels with a wide range of unique sequence variants (Fig. 6d), not merely identifying mutational varieties of a common conserved and active regulatory scaffold. Furthermore, we found that sequences spanning the whole gene regulatory structure improve the achievable dynamic range of gene expression compared to the common single regulatory regions[24] and shorter proximal[8,15] and core promoter parts[31] (Fig. 4a, b). Despite that in specific cases even single nucleotide variations can have a strong effect on expression[15,74], generally due to the natural evolutionary prerequisites of regulatory adaptability and robustness[75,76], short regulators simply cannot precisely control the full amount of expression in comparison to longer sequences spanning multiple regulatory regions, where regulatory adaptation is orchestrated across a range of meticulous and interacting sequence optimizations[3,9,77]. Consequently, the limitation and inability of current methods to design anything but short sequences spanning single regulators gives strong support for the use of generative approaches, capable of designing whole gene regulatory structures by learning from natural genomic and transcriptomic data directly, without requiring any screening experiments. A notable positive consequence of using transcriptomic data is also the ease of relating sequence to function, namely, expression levels and their dynamic ranges, whereas most alternative approaches focus primarily on protein expression via relative fluorescence intensity[8,15] or cell growth[24], making it harder to relate their measurements to a comprehensible gene expression scale (e.g. TPM) and thus potentially concealing more limited dynamic ranges than are initially understood. Moreover, despite using here the most relevant region sizes based on previously published results[3,7], we note that the functionality of the gene regulatory structure requires further extensive research in order to decipher key regulatory effects and interactions, while also validating the observations and applicability with different reporter genes, organisms, and tissues. Nevertheless, since recent optimized short sequence designs[15,24] are not capable of driving gene expression fully, with a large range of expression control remaining potentially untapped (Fig. 4d, f), the use of the whole gene regulatory structure offers a promising development focus for unlocking the full potential of gene expression control.

Generally, the properties and evolution of non-coding regulatory DNA are still poorly understood[3,78,79], which also makes interpreting generative DNA-sequence designs difficult. In contrast to proteins that carry conserved and structurally characterized protein domains[80], non-coding DNA sequences comprise many different cis-regulatory elements (Fig. 1a), including multiple binding sites for transcription factors[79], components of the transcription machinery[67,69] and chromatin remodeling proteins[3], as well as transcription initiation[52,53] and termination[27,54,65] related factors, which ultimately regulate and define the levels of gene expression. Recent studies have also shown that

other factors, such as weak motifs and interactions[8], motif associations across multiple regulatory regions[7], and DNA structural properties[10,81,82] are also strongly informative for gene expression predictions and might play an important role in expression regulation[3,8]. Therefore, it remains challenging to objectively define what constitues biologically feasible functional DNA, without relying on complex data-driven models for predictions[7,13]. Nevertheless, mining for the majority of the known, previously uncovered DNA regulatory grammar (Fig. 1a) showed that already with the initial non-optimized generator variant, generated sequences exhibit properties highly similar to those of natural regulatory DNA (Fig. 2), suggesting that adversarial network training sufficiently captures the regulatory sequence diversity present in natural DNA. Furthermore, coupling the generative and predictive models enables further rational design of regulatory DNA (Fig. 3a, c), as the generator is guided to operate within the feasible DNA sequence space learned by the predictor to produce functional DNA across the whole range of expression levels. This contrasts previous generative approaches with an arbitrary sampling of sequence spaces[11,12,32], instead, through model-based supervision, enabling controlled sequence design towards desired expression levels. The optimized ExpressionGAN-generated sequence variants are therefore found to carry known determinants of gene expression control (Fig. 5), reflecting the high and low expression-related properties expected based on previously published results[3,7,18,51,68,70]. Apart from correctly generating overall higher numbers of transcription activation-related DNA motifs and properties in highly-expressed genes compared to low ones, the generative model also learned to reproduce the relative positioning and co-occurrence of motifs, highly relevant for defining gene expression levels[3,19], across all cis-regulatory regions (Fig. 5). This demonstrates that our generative approach is well suited for the particular task of regulatory sequence design, inherently learning the general structure and features of regulatory DNA and mutating or combining them in novel yet functional ways.

By comparing the generative strategy of regulatory DNA design to a mutational one, we demonstrated experimentally that the generative approach is more suitable when using models trained purely on natural genomic data. Random mutagenesis-based approaches are commonly based on a similar brute-force strategy, starting with an existing natural sequence and traversing the sequence-expression landscape randomly, a set of mutations at a time, without considering the functional sequence context[15,32]. Since the computational and experimental screening processes that test the functionality of the designed sequences are decoupled from the sequence design stage[29], they require multiple isolated runs and experimental trials to progress beyond local minima and develop functional sequences[32]. This is the case even when using predictive models that can accurately map regulatory sequence to gene expression[7] (Fig. 6e–g). Moreover, since many biologically infeasible sequence variants are produced and can lead to untrustworthy predictions, they could be the reason behind the low success rate of the mutagenesis-produced sequence designs[33–35]. Apart from optimization, informing the mutational procedure by constraining the mutated positions to only relevant ones (Fig. 6e), such as those specified by the predictor that contains important binding sites[7], might also be an insufficient strategy to improve mutagenesis. This is potentially due to the large number of position-specific interactions for each single nucleotide position that affect protein-binding[22,83], which are spread beyond only the most important binding sites and their immediate vicinity[3,8,11]. For instance, constraining mutagenesis to the surrounding bases of the −35 and −10 promoter binding sites in E. coli led to producing very few functional variants[11,84], suggesting that the sequence beyond these regions contains important information for generating functional promoters. Similarly, we observed only a very small increase in the capacity to create sequences with increased or decreased expression levels when mutating only relevant positions (Fig. 6f). On the other hand, the generative approach

utilizes two knowledge-based models (Fig. 3a: both generator and predictor) and not only a single predictive model (Fig. 6e), where the sequence generator models the whole functional regulatory sequence landscape, producing natural-like functional sequence variants and not merely randomly mutated variants of existing sequences. Due to this, the generative approach explores the allowed sequence space, overcoming the drawback of exploring subspaces that contain infeasible sequence variants, such as when using random mutagenesis. Moreover, the problem of sequence-to-expression relevance is resolved by quantitatively mapping positional interactions across whole sequences, learning which positions are the most important for binding and functionality[11,12]. The generative strategy can thus produce fully functional DNA while deviating much farther from the known sequence space (natural level of ~33% or higher sequence divergence) compared to the mutational strategy, which shows limitations already with designs of at most 10% mutated sequence.

In synthetic biology, alternative bio-manufacturing hosts offer multiple benefits for speeding up bioprocess development[85] to facilitate high-yield manufacturing[86], bring new drug candidates to the clinic and maximize the use of manufacturing facilities during a pandemic[87]. To reach desired expression levels that are predictable, robust, and tunable, well-characterized gene regulatory parts are required for building genetic constructs[42]. When expressing a gene of interest, all regulatory regions have been shown to affect gene expression levels. For instance, the promoter can be strongly dependent on the choice of the terminator[7,88], and both are gene-context dependent and have to be matched with the coding region comprising an optimal codon usage to facilitate gene expression[89–91]. While there are tens of thousands of sequenced genomes, our capability to develop such advanced expression systems is highly underdeveloped[92] and primarily limited by the costly experimental screening approaches used to design and characterize short parts of single genomic regions[4,8,11]. This remains challenging for many industrially important strains due to low transformation efficiencies[93], as the screening techniques are intrinsically limited to organisms with high transformation efficiencies, targeting specific reporter genes under a particular biological condition[94–97]. Considering the costs and resource requirements of synthetic library construction and testing as well as potential benefits of generative modeling, the use and further development of mutagenesis for regulatory sequence design may not be worthwhile, apart from exploring the intrinsic functionality of expression regulation[8]. Instead, we demonstrate that the generative approach can produce regulatory DNA spanning the whole gene regulatory structure, while also considering information from the coding sequence, thus mimicking complete natural regulatory systems in order to ensure control over the full dynamic range of gene expression. The advantages of the proposed approach are that (i) it requires only natural genomic data as input, with no need for library construction and costly experimental screening, (ii) it could be expanded to any set of genes in virtually any sequenced organism, including those organisms with low transformation efficiencies, and (iii) it could be used to produce even condition-dependent models and regulatory sequences, including tissue/cell-type specific DNA designs with controllable gene expression levels. Therefore, we foresee this as a highly versatile and lucrative strategy to expand our knowledge of gene expression regulation as well as increase expression control in synthetic biology and metabolic engineering applications.

## Methods
### Data
*S. cerevisiae* S288C genome sequence data, including gene sequences, as well as transcript and open reading frame (ORF) boundaries, were obtained from the *Saccharomyces* Genome Database (https://www.yeastgenome.org/)[98,99] and additional published transcript and ORF

boundaries were used[100,101]. Coding and regulatory regions were extracted based on the transcript and ORF boundaries. DNA sequences were one-hot encoded, untranslated region (UTR) sequences were zero-padded up to the specified lengths (Fig. 1a: promoter of 400 bp, 5' UTR of 100 bp, 3' UTR of 250 bp, and terminator of 250 bp)[7] and the 64 codon frequencies were normalized to probabilities.

For gene expression levels, processed raw RNA sequencing Star counts were obtained from the Digital Expression Explorer V2 database (http://dee2.io/index.html)[59] and filtered for experiments that passed quality control, yielding 3025 high-quality RNA-Seq experiments. Raw mRNA data were transformed to transcripts per million (TPM) counts[102] and genes with zero mRNA output (TPM < 5) were removed. Prior to modeling, the mRNA counts were Box-Cox transformed[103] with lambda set to 0.22. As the mRNA counts and ORF lengths were significantly correlated due to the technical normalization bias from fragment-based transcript abundance estimation[104], we regressed out the gene length from mRNA counts. Specifically, the residual of a linear model, based on ORF lengths as the response variable and mRNA counts as the explanatory variable, was used as the corrected response variable, after which no correlation between gene length and its expression could be measured.

To obtain training datasets, we considered that for the initial 4975 protein-coding genes with genomic sequence information (predictor variables), median expression levels (response/target variable) across the RNA-Seq experiments varied within 1 relative standard deviation (RSD = $\sigma/\mu$) for 85% of the genes[7]. We, therefore, used DNA sequences of the regulatory and coding regions of these 4238 genes with RSD < 1 for training (Supplementary Fig. 1). For predictive modeling, the data comprised paired gene regulatory structure sequences as input variables and median mRNA counts as target/response variable, where a total of 3433 gene data instances were randomly selected for training the model, 381 for tuning the model hyperparameters and 424 for testing. For generative modeling, a total of 3814 regulatory structure sequences were used for training and the remaining 424 were used as unseen test data. Here, the data was balanced prior to training by distributing the corresponding mRNA counts across 30 bins and sampling input sequence data from all bins such that all the values were uniformly represented instead of using the initial distribution (Supplementary Fig. 23: the Box-Cox transformed data shown).

### Deep predictive modeling
To train a predictive model that predicts gene expression levels from whole gene regulatory structure data, the deep neural network architecture of 3 CNN layers and 2 dense (FC) layers was used[3,7,105,106]. The network was trained consecutively, first on regulatory sequences input to the first CNN layer and then the dense layers were replaced and the whole network retrained using the numeric variables (codon frequencies) appended to the output of the last CNN layer and input to the first dense layer. Batch normalization[107] and weight dropout[108] were applied after all layers and max-pooling[109] after CNN layers. The Adam optimizer[110] with mean squared error (MSE) loss function and ReLU activation function[111] with uniform[112] weight initialization were used. In total, 24 hyper-parameters (see initial value ranges in Supplementary Table 5) were optimized using a tree-structured Parzen estimators approach via the Hyperopt package v0.1.1[113] at default settings for 1500 iterations. The best models were chosen according to the minimal MSE on the validation set with the least spread between training and validation sets. The coefficient of determination ($R^2$) was defined as $R^2 = 1 - \mathrm{SS}_{\mathrm{Residual}}/\mathrm{SS}_{\mathrm{Total}}$ [Eq. 1], where $\mathrm{SS}_{\mathrm{Residual}}$ is the sum of residual squares of predictions and $\mathrm{SS}_{\mathrm{Total}}$ is the total sum of squares, and statistical significance was evaluated using the two-tailed $F$-test. For training deep models and data collection, Tensorflow v1.12.0 and Keras v2.2.0 software packages were used and accessed using the Python interface.

## Deep generative modeling

To devise a system to generate realistic DNA regulatory sequences corresponding to the whole gene regulatory structure, we trained a generative model using a generative adversarial network (GAN) approach[46] (Fig. 1d, Supplementary Fig. 1). In order to capture all the levels of regulatory information across the input sequences, both the generator and discriminator equally comprised 6 convolutional neural network (CNN) layers of opposite orientation, where the first (last) 5 layers were residual blocks containing skip connections with a residual factor of 0.3[17,114]. Each CNN layer comprised 100 filters, a kernel size of 5 and a stride of 1. The dense layer size was equal to the input sequence size (1000) × CNN filter size (100). The Adam optimizer[110] with the Wasserstein loss function (WGAN)[115,116] and ReLU activation function[111] with uniform[112] weight initialization were used. The learning rate parameter was set to 1e−5, beta1 to 0.5 and beta2 to 0.9, and the batch size was 64. The ratio of the discriminator to generator updates was set to 5. The dimensionality of the latent space was set to 200 after testing GANs with 100, 200, and 1000 dimensional latent spaces and finding no improvement in performance over this size, showing that it sufficiently captured the key information in the DNA sequence data. For single region generators, the exception was with models with a sequence size smaller than 200 bp (5′ UTR, proximal and core promoter generators), where the dimensionality of the latent space was set to 100. The latent space was sampled according to a standard normal distribution during training.

To generate sequences that manifest desired target expression levels by connecting the functional regulatory DNA space modeled by the generator with expression levels and coding sequence information modeled by the predictor, a DNA-based activation maximization approach[17,73] was used that incorporates both the trained generator and predictor models (Fig. 2a). This approach has been tested with generative modeling of both regulatory DNA[17,32] as well as in other domains, such as image modeling, where its demonstration equals that of biological experimental validation[57,73] due to human-level performance being the benchmark there. The optimal trained generative model to use for optimization was identified at iteration 200,000 (Fig. 1c, Supplementary Fig. 2), further supported by comparing the properties of generators obtained at 6 different training iteration checkpoints (100,000, 200,000, 300,000, 500,000, 700,000 and 1,000,000) after optimization, which included the percentage of unique generated sequences, range of predicted gene expression levels and amounts of sampled sequences across the whole expression range (Supplementary Fig. 24). Optimizations were run for 100,000 iterations and, to increase the breadth of the investigated latent subspace, 10 optimization runs were performed with different initial random states. The results were merged to obtain a set of 6,062,804 unique sequences that were used for further analysis.

To obtain a selection of sequences for experimental validation, the following selection procedure was used. Four expression bins were defined to cover a 4 order of magnitude range of expression levels within a 10% range above or below the TPM values of 10, 100, 1000, and 10,000. Approximately 100 sequences per expression bin and per optimization seed were randomly selected from the above merged optimized sequence dataset, yielding 5706 sequences. Next, by comparing 16 sequence properties (see underlined properties in Supplementary Table 1) of the generated sequence selection to those of natural test sequences, 452 sequences were sub-selected with all tested sequence properties within the ranges defined by natural test sequences. From here, the experimental set was constructed by randomly selecting 10 sequences in each expression bin, by optimizing for the highest sequence diversity within each expression bin, whilst retaining the natural sequence diversity (avg. seq. id. -0.67). The final set of 40 selected sequences was thus highly diverse and as different from natural sequences as these are among themselves, representing as yet unseen sequence variants. Further limitations with sequence synthesis when ordering the selected generated variants as gene fragments from either TWIST Bioscience (www.twistbioscience.com) or IDT (gBlocks, https://eu.idtdna.com/) (Supplementary Table 6) resulted in the final experimental set of 17 sequences (Supplementary Table 7), with 4 from the expression bin of -10 TPM, 6 from -100 TPM and 7 from -1000 TPM (Supplementary Table 2).

## Mutagenesis approach

To design regulatory sequence variants with the mutation procedure, promoter sequences were randomly mutated at different settings for the percentage of the mutated sequence size: 1% (4 bp), 2% (8 bp), 5% (20 bp), and 10% (40 bp). This was done while verifying that all mutated variants were different from any of the natural sequences, thus the mutation size also corresponded to the distance from the closest natural sequence. The maximum mutation size of 10% was used in order to limit using the predictor too far outside of its operational range, defined by the natural training sequence space, which can potentially cause incorrect predictions[34,117]. 300,000 mutations were performed per each of the eight settings per gene scaffold sequence.

Either whole promoter sequences of 400 bp or only the most relevant positions were mutated. For the latter, the predictor was used to inform the mutational procedure by querying its sensitivity to specific positions in the promoter sequence (Fig. 6e), so that only the most sensitive and thus relevant positions were preferentially used as a guide for targeted mutagenesis (Supplementary Fig. 20). To calculate the relevance of the different DNA positions for model predictions, defined as Relevance $= (Y - Y_{occluded})/Y$ [Eq. 2], where $Y$ is the model prediction, an input dataset with sliding window occlusions was used with the predictive model to obtain predictions[118,119] (Supplementary Fig. 25). The window size of the occlusions was set to either 1 or 10 bp. To obtain only highly sensitive regions, relevance z-scores above a cutoff of 1 were selected.

To calculate the number of mutated sequence variants that achieved an over 50% increase or decrease in predicted gene expression level, regulatory sequence scaffolds from the following 7 genes were used: YDR541C, POP6, PMU1, YBL036C, MNN9, RPC40, RPL3. Experimental sequence selection was performed with the RPL3 gene, where the mutated sequences were sorted and selected based on the largest achieved increases and decreases, targeting ~2-fold changes, as well as according to the limitations imposed by DNA sequence manufacturers. Thus, only sequences with a mutated sequence size of 5 and 10% were selected, where either the whole promoter or only relevant positions with a window size of 10 bp were mutated. When selecting for increased expression while mutating whole promoters, only sequences with a mutated size of 10% achieved the targeted changes in predicted expression levels. Two representatives were selected for increased gene expression per combination of settings and a single representative for decreased expression, yielding the 10 tested sequence constructs (Supplementary Table 4).

## Experimental strain construction

The *S. cerevisiae* strain S288C (ATCC no. 204508) was used as the base strain for all genetic engineering. Promoter (including 5′ UTR) and terminator (including 3′ UTR) DNA sequences were ordered as gene fragments from either TWIST Bioscience (www.twistbioscience.com) or IDT (https://eu.idtdna.com/). The exception was the RPL3 promoter and RPL3 terminator, for which fragments could not be synthesized due to sequence complexities, and were thus amplified from the genome with promoter_YOR063W_fwd, promoter_YOR063W_rev, and terminator_YOR063W_fwd, terminator_YOR063W_rev primer pairs, respectively (Supplementary Table 8). For the promoter-GFP-terminator constructs, the UBIMΔkGFP* version of the GFP gene from Houser et al.[120] was used (Supplementary Table 9).

Integration of the promoter-GFP-terminator constructs into the genome at the XI-2 locus was performed using the CRISPR/Cas9

plasmid (pCFB2312) and gRNA helper vectors (pCFB3044) from the EasyClone marker-free system[121]. Transformations into the S288C strain were performed via the following procedure. The S288C strain harboring pCFB2312 was cultivated in YPD medium consisting of 10 g/L yeast extract (Cat#Y1625, Merck), 20 g/L peptone (Cat#1072241000, Merck), 20 g/L glucose (Cat#1083422500, Merck) supplemented with 200 mg/L G418 (Cat#10131035, Thermo Scientific) and grown into competent cells freshly at OD600 1.3-1.5. 300 ng pCFB3044 and 1 µg fragments were transformed into the competent cells by the standard lithium acetate (LiAc)/single-stranded DNA (SS-DNA)/polyethylene glycol (PEG) method[122] and plated on YPD agar plates containing 200 mg/L G418 and 100 mg/L nourseothricin (Cat#AB-102, Jena Bioscience GmbH).

Promoter-GFP-terminator fragments were prepared by ligating three fragments: the promoter with 90 bp overlap to the genome and 90 bp overlap to the GFP gene, the GFP gene, and the terminator with 90 bp overlap to the GFP gene and 90 bp overlap to the genome. The exception was the RPL3 promoter and terminator, which were amplified from the S288C genome with a shorter 40 bp overlap flanking the primers. Fragments were prepared by ligating the promoter, GFP gene, and terminator fragments together with a linearized pUC19 plasmid (Cat#OGS590, Merck) by Gibson assembly[123], for which 5 µL of fragment and vector mixture was mixed with 5 µL Gibson Mastermix (Cat#E2611, NEB) and incubated in 50 °C for 1 h before transforming into *E. coli* DH5α competent cells. The pUC19 vector was linearized by PCR with the pUC19_fwd and pUC19_rev primer pair (Supplementary Table 8), with 20 bp overlaps flanking the ends for the Gibson assembly. All plasmids were extracted using the GeneJET Plasmid Miniprep Kit (Cat#K0502, Thermo Scientific) and used as the templates for the promoter-GFP-terminator fragments with the L90 and R90 primer pair (Supplementary Table 8). To obtain strains with correctly integrated fragments at the XI-2 locus, colonies were verified with PCR using the 909[121], GFP_rev and 910[121], GFP_fwd primer pairs (Supplementary Table 8) and the fragments were sequence-verified at Eurofins Genomics (https://eurofinsgenomics.eu/) after amplifying them with the L90, R90 primer pair (Supplementary Table 8).

For the mutagenesis experiment, designed promoter_RPL3 variants were ligated with GFP and the native terminator_RPL3 with the methods described above (Supplementary Table 9). For the generative experiment, the different generated synthetic promoters and terminators corresponding to 17 whole constructs were ligated with GFP with the methods described above (Supplementary Table 7). PCR reactions were carried out with Phusion High-Fidelity Polymerase (Cat#F-530, Thermo Scientific) and gel-purified with GeneJET Gel Extraction Kit (Cat#K0691, Thermo Scientific). Primers were designed using the Primer3 software (https://benchling.com) and synthesized by IDT (https://eu.idtdna.com/).

## RNA extraction and quantitative PCR

All yeast strains were cultured and monitored in a 48-well FlowerPlate (m2p-laboratories GmbH, Germany) at 30 °C and 1200 rpm using a microbioreactor Biolector (m2p-laboratories GmbH, Germany). Cultures were started from a preculture grown overnight, at an OD600 of 0.03 in 1 mL minimal media with 2% glucose (Supplementary Table 10). OD600 was monitored in real-time by the Biolector approximately every 20 min. After 15 h of cultivation, when the cells were in a mid-exponential growth phase, the cells were collected and immediately used for RNA extraction with the RNeasy Mini Kit (Cat#74104, Qiagen). For each batch of cultivation, the S288C wild-type strain, as well as the two integration strains with the POP6 and RPL3 regulatory regions, were used as control groups. All cultivations were performed in biological triplicates.

cDNA was synthesized with QuantiTect Reverse Transcription Kit (Cat#205311, Qiagen) by adding 50 ng of total RNA to a final RT reaction volume of 20 µL. 1 µL of the cDNA was used as a template with the Thermo Scientific DyNAmo Flash SYBR Green PCR Master Mix in a Mx3005P QPCR System (Agilent Technologies, USA). A 2 step qPCR protocol was used: 10 min initialization at 95 °C and 40 cycles of each: 30 s 95 °C and 60 s 60 °C. *S. cerevisiae* TAF10[124] was selected as the reference gene and the qPCR primer pair for TAF10 was used (Supplementary Table 8: TAF10_qPCR _fwd and TAF10_qPCR _rev). qPCR primers for GFP (Supplementary Table 8: GFP_qPCR_fwd and GFP_qPCR_rev) were designed using IDT's PrimerQuest tool v2.2 (https://eu.idtdna.com/pages/tools/primerquest). The qPCR primers were synthesized by IDT (https://eu.idtdna.com/).

Measurements were performed in separate batches due to the constraints of the measurement plate size of 96 wells. For each qPCR batch, samples from the S288C wild-type strain, as well as the two integration strains with the POP6 and RPL3 regulatory regions, were included as the respective reference, low-expression, and high-expression control groups. Each sample has technical duplicates. Cycle thresholds (Ct) of the reporter gene were normalized relative to the Ct value of TAF10[124]. The $2^{-\Delta\Delta CT}$ (avg. 2pddct) value was used as the indicator of the relative expression level of GFP for each construct[125], where the wild-type strain was used as the reference (Supplementary Table 2 and S4). The values were equalized across all qPCR batches based on the known TPM values of the native POP6 and RPL3 controls present in every batch, using a linear curve fit to infer the TPM values of each replicate of the generated constructs.

## Data analysis and software

The performance of the generative model was monitored by measuring the sequence properties of the generated variants, including (i) sequence compositional validity, (ii) sequence similarity measures, (iii) predicted gene expression levels, and (iv) known *cis*-regulatory grammar (Supplementary Table 1), and by testing, if they reflected the properties of natural sequences. DNA sequence homology was calculated with the *ratio* function in the python-Levenshtein package v0.12.2, equaling the Levenshtein (edit) distance divided by the length of the sequence. The Jaccard distance between two DNA sequences was defined as the intersection over the union of sets of their unique k-mers of size 4. Known motifs were located based on the shortest edit distance between sequence segments and the motif, using the *distance* function in the python-Levenshtein package v0.12.2, except for deep learning-uncovered motifs and motifs associations[7] for which the *partial_ratio* function from the fuzzy-wuzzy package v0.18.0 was used, and TFBS for which the *fimo*[49] and *tomtom*[126] functions from Meme suite v5.0.2[127] were used. Nucleosome depletion was calculated using the R package nuCpos v3.8[55,56]. Samples of 64 generated or natural test sequences were used per parameter except where stated otherwise.

For analysis of high and low expression bins, 20,000 generated sequences were sampled from ExpressionGAN after 50,000 optimizer iterations and verified to be valid (Supplementary Table 1) and unique to any natural sequence. The high expression bin was defined as 1e2 < TPM < 1e5 and the low expression bin as 1e0 < TPM < 1e2, where the upper and lower limits were used to remove outliers (Fig. 5a). The T-rich and T-poor motifs included 5′-['TTTT', 'TTCT', 'GTTC', 'CTTT', 'TTC', 'CTTA', 'TCTA']−3′ and 5′-['AGGA', 'AC', 'GAGC', 'AGCA', 'ACGG', 'AAGA', 'AGCG']−3′, respectively, as defined in previous studies[51,68].

Python v3.6 (www.python.org) and R v3.6 (www.r-project.org) were used for computations. For statistical hypothesis testing, Scipy[128] v1.5.4 was used with default settings. All statistical tests were two-tailed except where stated otherwise. For correlation analysis, the Spearman correlation coefficient is reported and statistical significance was tested using a *T*-test. For data analysis, Tensorflow v2.4.1, Pandas v1.1.5, Scikit−learn v0.24.2, Scikit-bio v0.5.6, Biopython v1.78, Tidyverse v1.3.0, and packages were used. Seaborn v0.11.1 was used for visualization.

## Reporting summary

Further information on research design is available in the Nature Research Reporting Summary linked to this article.

## Data availability

Genomic data, transcript and gene boundaries used in this study were obtained from the *Saccharomyces* Genome Database (https://www.yeastgenome.org/) and Ensembl (https://www.ensembl.org/), RNA sequencing data from the Digital Expression Explorer V2 database (http://dee2.io/mx/), DNA sequence motifs from the Meme suite motifs databases file (http://meme-suite.org/) and additional data from the cited references (links to raw data in Supplementary Table 11). Sequence data generated in this study are provided in Supplementary Tables 6, 7, and 9, and experimental data in Supplementary Tables 2 and 4. Source data were deposited to the Zenodo repository and are available at https://doi.org/10.5281/zenodo.6811225.

## Code availability

Code was deposited to the Github repository and is available at https://github.com/JanZrimec/ExpressionGAN.

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

## Acknowledgements

We thank Filip Buric and Sandra Viknander for technical discussions and critical comments as well as Benjamin Heineike, Kate Cambell, and Simran Aulakh for proofreading and providing critical feedback on the manuscript. We gratefully acknowledge the NVIDIA Corporation for supporting this research as well as the Chalmers Center for Computational Science and Engineering (C3SE) and the Swedish National Infrastructure for Computing (SNIC) for providing computational resources. Mikael Öhman and Thomas Svedberg at C3SE are acknowledged for technical assistance. The study was supported by SciLifeLab funding (A.Z.), Swedish Research council (Vetenskapsrådet) starting grant no. 2019-05356 (A.Z.), BigData@Chalmers funding initiative (Area of Advance ICT) (A.Z.), Marius Jakulis Jason foundation (A.Z.), Slovenian Research Agency (ARRS) grant no. J2-3060 (J.Z.), Public Scholarship, Development, Disability, and Maintenance Fund of the Republic of Slovenia grant no. 11013-9/2021-2 (JZ) and EU Horizon 2020 research and innovation program under the Marie Skłodowska-Curie grant agreement no. 722 287 (C.S.B.). Computing resources at the Chalmers Center for Computational Science and Engineering (C3SE) were partially funded by the Swedish Research Council through grant agreement no. 2018-05973 (A.Z.).

## Author contributions

J.Z. and A.Z. conceptualized the project; J.Z., A.S.M., N.S., V.V., M.H.C., D.D., and A.Z. designed the computational analysis; J.Z., A.S.M., and N.S. performed the computational analysis; J.Z., X.F., C.S., V.J., C.S.B., V.S., F.D., and A.Z. designed the experimental analysis; X.F., C.S. and V.J. performed the experimental analysis; J.Z. and A.Z. interpreted the results; J.Z. and A.Z. wrote the draft; All authors contributed to the final manuscript.

## Funding

## Competing interests

The authors declare no competing interests.
