## [Peer Review File · Nature Communications]

Reviewers' Comments:

Reviewer #1:

Remarks to the Author:

The authors apply a GAN toward native sequences of regulatory elements in an effort to identify the function of synthetic sequences. At the onset, this work appears to be of interest in the field. As the paper is being read, there is a lack of transparency as to (1) how this work compares with others in the field that have used both adversarial GANs as well as other approaches to do something similar and (2) what was actually achieved in this work esp. as it relates to Figure 4. For the later, I found this section very hard to unpack and see what was done. These figures look very similar to the prior ones whereas specific constructs were made here and a better representation of the data should be provided. Where is the proof that these synthetic operators actually perform? For the former, it would be good to see the results of Figure 4 contrasted with results of other computational approaches. How does this ExpressionGAN compare with other methods to do the same? How is the knowledge gained any different than what others have proposed previously? These overall metrics and comparisons are critical for understanding the impact of this work.

Reviewer #2:

Remarks to the Author:

In this work, Zrimec et al. introduce a method for designing regulatory non-coding DNA sequences in yeast using deep neural networks. Specifically, the authors train a Generative Adversarial Network (GAN) on published genomic and RNA-seq data with the purpose of generating novel regulatory sequences that contain endogenous-like features. The GAN learns to simultaneously generate promoter, UTR, and terminator sequences. Furthermore, they combine the GAN's generator network with a previously developed predictive model of gene expression to design sequences with specified performances. Finally, they experimentally test a small subset of designed promoters and find multiple sequences that result in stronger expression compared to highly expressed natural controls.

Overall, the work presented is a positive contribution to the synthetic biology literature, where deep neural networks trained on large publicly available datasets are increasingly being used for sequence design. It is noteworthy that the GAN can be successfully trained on endogenous data only. However, given that the task is to design endogenous-like sequences, perhaps this result is not that surprising. Furthermore, the authors do a comprehensive job at ensuring that the designed sequences contain native-like features while being different from endogenous sequences and from one another. Finally, the approach of connecting the GAN's generator to a predictor and optimizing in the latent space of the generator is simple yet effective.

However, there are a few issues that I think need to be addressed before publication.

- The introduction frames the present study as an improvement over the traditional method of "experimental screening of large amounts of random synthetic sequences" (lines 64-65). This is written as to imply that most sequence design efforts have consisted of little more than testing fully random sequences chosen without any direction. In reality, the first decade of synthetic biology was dominated by the moderately successful rational design approach, where regulatory sequences were handcrafted and tested in low-throughput assays. A bit further on the authors also write (line 114-115): "we aimed to improve upon random mutagenesis." In reality, there is a rich literature of papers that use generative models to design functional sequences. I strongly recommend the authors modify their introduction accordingly and add relevant references.
- It is very difficult to understand which regulatory sequence regions the generator model is actually trained on without reading the methods section. For example, lines 122 and 123 contain the vague descriptor "whole gene regulatory structures", which I recommend should be replaced with the actual sequence regions (promoter, UTRs, terminator). Another set of vague descriptors are "variables describing the coding region" (lines 202-203) and "coding region information" (line 226), which should be replaced by "codon frequencies" if that is the only feature. Furthermore, Figure 1A needs to be fully reworked so that it clearly indicates this information. The current thin

dashed lines below the regulatory regions are hard to notice, and the figure gives the overall impression that even CDS sequences are generated.

- There is a lack of baselines against which the presented method is evaluated. It is not clear why the authors did not compare to approaches other than random mutagenesis such as genetic algorithms proper and in particular simulated annealing. In addition, while we don't expect the authors to run a full comparison against every model-based or generative neural network approach ever developed, they should more clearly acknowledge the existence of that literature.
- The paper spends a lot of time dissecting the features of the generated sequences within each individual regulatory element, with some interesting + detailed analysis. It would have been interesting to see some kind of ablation study here, e.g. training a generator on the leave-one-out permutations of the promoter, 5'-UTR, 3'-UTR, terminator. This would make a more convincing argument on whether using the entire regulatory ecosystem for deep learning-based sequence design offers a real advantage; and would also lend the individual regulatory analysis breakdown more impact. Alternatively, it might suggest e.g. some regulatory regions have higher impact or are easier targets for design/have wider dynamic range, which would also be interesting.
- Closely related to the previous point: one of the purported advantages of the authors' design method is the ability to simultaneously design promoter, UTR, and terminator sequences to optimize expression. However, the extent to which this is an improvement over separately optimizing them is not explored. The following analysis could help address this issue: 1) Train separate GANs and predictors on each regulatory region. 2) Use these separate networks to design regulatory elements in isolation to maximize expression. 3) Use the combined predictive model to determine whether these elements in combination would lead to higher expression.
- Please include the 24 designed sequences that could not be synthesized by either twist or IDT?
- The authors mention in several places that their method allows designing gene expression across 6 orders of magnitude (e.g. line 459), yet their experimental results show coverage of little more than two (Figure 5D). Please clarify.
- Minor point: can the authors elaborate on why they find positive correlation between the number of adenines around the start codon and predicted expression levels of generated sequences (lines 261-262)? The authors seem to be making the connection to Kozak sequences (line 259). However, Kozak sequences regulate translation, but the training data for the models in this study is genomic and RNA-seq information.
- The following figure references might be incorrect: Figure 2G (line 365) and Figure 2F (line 371).

Replies to reviewer comments

Reviewer #1 (Remarks to the Author):

Comment 1.1

The authors apply a GAN toward native sequences of regulatory elements in an effort to identify the function of synthetic sequences. At the onset, this work appears to be of interest in the field.

Reply 1.1

We thank the reviewer for his positive and constructive comments, which we have addressed in full in the revised manuscript, with answers and explanations provided below.

Comment 1.2

As the paper is being read, there is a lack of transparency as to (1) how this work compares with others in the field that have used both adversarial GANs as well as other approaches to do something similar

Reply 1.2

We have worked to resolve the issue of transparency toward existing similar work and comparison with such work throughout the paper. To make our approach and results more clear and accessible to a wider audience, we have reshaped and expanded the introduction section (lines 41-124, pages 3-5), adding additional overview of recent approaches both from the mutational and generative fields to explain state of the art approaches and the potential benefits of using generative modeling. In total, 30 new references were added to the manuscript in this revision. The key focus in the introduction was to pinpoint and clearly explain the limitations of both types of approaches and how the present study goes beyond these limitations, trying to resolve them. With mutagenesis, this includes producing and exploring sequences with arbitrary random mutations in all the related approaches including genetic algorithms [1,2]. Since these sequence candidates can be biologically infeasible, it leads to untrustworthy predictions and resource inefficiency with multiple testing and design rounds [2–5] (lines 59-78, pages 3-4).

Specifically, genetic algorithms and their derivatives also rely on random mutagenesis, introducing arbitrary mutations in every round of candidate generation, since they do not explicitly model the allowed sequence-function landscape in the sequence design step [1,6–9]. Rather, they rely fully on the computational screening step (by predictive models, also termed oracles) to understand sequence functionality and to guide sequence design [1,2]. However, predictive models are highly sensitive to sequence validity and displaying worsening performance with sequence deviations from the training data, especially when traversing towards biologically infeasible sequence subspaces. Since also infeasible sequences are produced in the random mutagenesis step, testing such candidates can stress the computational screening methods and lead them into pathological states, where predictions are highly untrustworthy [3–5]. Apart from this, only a small subset of sequences actually conform to the target design principles, and finding them is hard among the enormous sequence space, with many local minima that these algorithms can get stuck in [2]. Therefore, present mutational approaches are highly resource inefficient, as they require many separate runs to explore different settings, generally requiring unnecessary testing of many infeasible sequences as well as experiments to verify the functionality of the designed sequence sets. On the other hand, due to being capable of modeling the natural sequence-function landscape, generative approaches can directly draw biologically feasible sequence samples.

However, current approaches are limited to screening and designing DNA from individual regulatory sequence parts, even in studies where generative modeling is used [3,10,11]. The studies demonstrating the use of generative models to design regulatory DNA also display other drawbacks, such as not including gene expression optimization [10], not performing experimental validation, testing polyadenylation but not transcription design capability

specifically [3], or focusing on genes but not regulatory regions [12] (lines 80-101, page 4). On the other hand, in our previous work, we demonstrated that regulatory information is encoded in the whole gene regulatory structure, including the promoter, UTRs, terminator, and coding regions [13,14]. For instance, we experimentally showed that while using strong native promoters it is possible to predictively increase/decrease mRNA levels by only changing terminator sequences. This was the motivation to develop the ExpressionGAN approach that is capable of generating an entire regulatory sequence that is matched to the coding region. By including whole gene regulatory structure and experimentally verifying the design approach, we expand current knowledge and capabilities and provide a highly useful advancement in the field.

In the revised manuscript version, we tried to clarify the transparency of this work in comparison with existing approaches and knowledge in the field as much as possible, across all sections of the manuscript. We have made further detailed explanations of all the revisions and especially how they relate to improving the transparency in the following replies.

Comment 1.3

and (2) what was actually achieved in this work esp. as it relates to Figure 4. For the later, I found this section very hard to unpack and see what was done. These figures look very similar to the prior ones whereas specific constructs were made here and a better representation of the data should be provided.

Reply 1.3

The ExpressionGAN devised in the present study is based on existing thoroughly validated published methods, which include the following. (i) Generative adversarial networks (GAN) [15], which have been demonstrated as the top generative modeling approach for producing optimal sequence-function mappings [3,10,11,16,17]. (ii) An activation-maximization optimization [2,18–20] approach, which, due to multiple tests in the vision domain with human performance verification, has had its performance tested at a level equal to experimental validation [5,19,20]. (iii) Sequences spanning the whole gene regulatory structure [14] (Figure 1A) and (iv) highly-accurate predictive models of gene expression levels that can explain over 82% of expression variation from regulatory sequence alone (Figure 1B), both tested in a previous study [13]. This is detailed in the second paragraph of the discussion section (lines 550-576, pages 21-22).

We further test and validate this generative approach in numerous ways in separate controlled experiments, demonstrating its performance and usefulness, including the following comparisons. (i) Comparing the properties of generated sequences to natural ones (Figure 2). (ii) Comparing the use of the whole gene regulatory structure to single regulatory regions and shorter promoter parts, which are commonly used with both generative approaches [10] and mutagenesis [6,8,9] (Figure 4). (iii) Testing whether generated sequences contain known DNA regulatory grammar properties that drive gene expression to low and high values (Figure 5). (iv) experimental demonstration and validation of ExpressionGAN-generated sequences (Figure 6A-D), designed specifically to test as different sequences as possible (both to natural and among themselves, Figure 6A,B) to properly validate the approach. (v) Further experimentally contrasting the generative approach with a standard mutational one that does not inherently model the allowed sequence-function landscape (Figure 6E-F). The above comparisons are further explained and discussed in the third and following paragraphs of the discussion section (lines 578-614, pages 22-23).

Regarding the Figure 4 (presently renamed as Figure 5), we agree with the reviewer that the section has very dense material and we also revised this chapter of the results (lines 271-337, pages 11-13) by inviting 3 separate native or well trained English speaking researchers to read and comment on the whole manuscript, to make sure that it is clear for the readers. Additionally, we tested all reported values with statistical tests, where possible, to ensure proper statistical stringency and reporting only on significant values, though unfortunately this contributed to the density of the text. We also tried to further aid the analysis with Figure 5C, which gives an overview of the results and findings in this section.

While in Figure 2 (chapter 2 of results section, lines 183-221, pages 8-9), we analyze the DNA regulatory grammar carried in generated sequences and compare it with the grammar present in natural sequences, in Figure 5, the generated sequences are segregated into groups with overall 'high' or 'low' expression levels (Figure 5A) and we explore more specific DNA regulatory properties that are known to drive gene expression, determining if they correspond to the overall predicted expression levels of the sequences. We believe that this section is very crucial for the manuscript as it aggregates a large part of the knowledge of *cis*-regulatory sequence grammar and demonstrates that ExpressionGAN recapitulates known DNA patterns that are important for mRNA level control in its sequence designs (please see overview in Figure 5C). Briefly, non-coding DNA sequences comprise many different *cis*-regulatory elements (Figure 1A), including multiple binding sites for transcription factors [21], components of the transcription machinery [22,23] and chromatin remodeling proteins [14], as well as transcription initiation [24,25] and termination [26–28] related factors, which ultimately regulate and define the levels of gene expression. Recent studies have also shown that other factors, such as weak motifs and interactions [29], motif associations across multiple regulatory regions [13] and DNA structural properties [30–32] are also strongly informative for gene expression predictions and might play an important role in expression regulation [14,29]. It is thus challenging to objectively define what is a 'proper' functional DNA just by looking at the sequence alone. Nevertheless, we explored sequences generated by our models for the majority of the known, previously uncovered DNA regulatory grammar (Figure 1A). We found that already with the initial non-optimized generator variant, generated sequences exhibit properties highly similar to those of natural regulatory DNA (Figure 2), suggesting that adversarial network training sufficiently captures the regulatory sequence diversity present in natural DNA. Furthermore, coupling the generative and predictive models enables further rational design of regulatory DNA (Figure 3), as the generator is guided to operate within the feasible DNA sequence space 'understood' by the predictor to produce correct functional DNA across the whole range of expression levels. The optimized ExpressionGAN-generated sequence variants are therefore found to carry known determinants of gene expression control (Figure 5), reflecting the high and low expression-related properties expected based on previous published results [13,14,33–36].

Therefore the present Figure 5 and fifth chapter of results (lines 271-337, pages 11-13) go beyond what is shown in previous parts, namely, chapter and Figure 2, as explained below. In chapter and Figure 2 (lines 183-221, pages 8-9), the initial trained generative model was used, to verify its general capability of producing sequences with overall distributions of known DNA regulatory grammar and if they are similar to those of the natural data that was used to train the model. However, despite generating DNA across a range of expression levels, the initial generator was not further optimized to truly span the whole possible dynamic range of gene expression and properly generate sequences with increased or decreased expression levels, as it enables only random sampling of sequences in relation to their expression levels. On the other hand, with further development, using the optimization procedure as described in chapter 3 of the results (lines 223-270, pages 10-11, Figure 3A), the optimized ExpressionGAN enables directly sampling sequences with desired expression levels, on demand. Therefore, in chapter 5 we then tested these specific capabilities of the optimized generative procedure (ExpressionGAN), by testing large batches of samples of both sequences optimized for low expression as well as high expression (Figure 5A: 10,000 samples each). As is detailed in the text, we tested for the presence of regulatory motifs and

properties both across the whole gene regulatory structure, as well as focused more specifically on the crucial properties in the core promoter region (Figure 5C), finding that ExpressionGAN can recreate both low and high expression-related properties across the whole gene regulatory structure in accordance with existing knowledge. This gave us the green light to go ahead with experimental validation of the method - for more details on this, please see Reply 1.4 below.

We hope that we have clarified the content and specifics of this section and that the reviewer will find it suitable for publication.

Comment 1.4

Where is the proof that these synthetic operators actually perform?

Reply 1.4

In our study, we provide multiple separate computational and experimental analyses together providing proof that our generative approach (ExpressionGAN) generates fully functional regulatory DNA designs that perform *in vivo* within their desired expression level targets. After training the initial generative model (Figure 1, results lines 127-182, pages 6-7), we gained control over the generator's mapping of the regulatory sequence space to the respective expression levels and navigated the functional regulatory sequence-expression landscape, to produce generated sequences with desired expression levels in a range of almost 6 orders of magnitude of TPM according to computational predictions (Figures 3 and 4A,B, results lines 223-270, pages 10-11). We performed in-depth computational analyses showing that the newly generated DNA designs contain natural-like properties and *cis*-regulatory grammar (Figure 2, results lines 183-221, pages 8-9) that is known to regulate gene expression levels [13,14,33–36] (Figure 5, results lines 271-337, pages 11-13) - please see also Reply 1.3 for more details.

Then, as detailed in chapter 6 of the results section (lines 433-537, pages 17-20), by testing sequence designs generated by the generative models, we found that the sequences could achieve the overall predicted expression levels (Figure 6D, Figure S17, Spearman's $\rho = 0.74$, p -value $< 1.6e-14$) and 57% of them surpassed the naturally highly-expressed controls. This was achieved despite preferentially selecting sequences with the lowest sequence similarity to natural ones (Figure 6A, Figure S14: $>30\%$ difference in avg. sequence identity) and maximizing sequence diversity in each expression range (Figure 6B, Figure S14), meaning that the sequences were completely novel and unlike any other natural DNA. The experimental analysis was designed specifically to test the most divergent possible sequence variants, with an average sequence identity well below 70% both to natural sequences and amongst themselves (Figure 6A,B). This validated the approach *in vivo* across 3 orders of magnitude of expression levels with a wide range of unique sequence variants (Figure 6D), not merely identifying mutational varieties of a common conserved and active regulatory scaffold. Despite this, the sequences were feasible and properly recognized by the predictor (Spearman's $\rho = 0.74$, p -value $< 1.6e-14$). To increase clarity of these results in the manuscript, apart from overall proofreading and revisions to the text in chapter 6 of the results section (lines 433-537, pages 17-20), we have also expanded and revised the discussion section to explain the results (e.g. lines 578-614, pages 22-23), putting them in a more general validation context and relaying the information more clearly.

Comment 1.5

For the former, it would be good to see the results of Figure 4 contrasted with results of other computational approaches. How does this ExpressionGAN compare with other methods to do the same?

Reply 1.5

Indeed, we fully agree with the reviewer's suggestion and have prepared a whole new set of analyses that are presented in chapter 4 of the results section (lines 271-337, pages 11-13, Figure 4). Here, we compared our work to the current existing state of the art solutions, which are all based on single regulatory regions or parts of these [6,8–10] (please see also Reply 1.2 above), in order to assess whether using the whole gene regulatory structure outperforms these single-region solutions. Continuing with the explanation in the above Reply 1.2, already in previous studies, including our own recent one [13], it was shown that regulatory regions each contribute a certain amount of information for predicting gene expression levels [9,28,37–40], with each region carrying a given amount of overlapping as well as unique information, thus jointly contributing to gene expression variation [13,14]. In fact, whereas the initiating 5' regions (promoters and 5' UTRs) seem to define large scale gene expression properties (turning expression on/off), the role of terminating 3' regions (including 3' UTRs and terminators) is to fine-tune expression levels [14,41]. Hence, varying a regulatory region while keeping others intact can lead to a large measurable degree of regulatory freedom per gene [13]. Additionally, from the technical point of view, only the whole gene regulatory structure constrains the functional sequence space to the biologically feasible regulatory variants. Even when sampling only a single region, it must still be based on the knowledge of the effect of the whole surrounding regulatory structure, as this is the primary state of natural regulatory systems [13,14,41] according to which also experimental systems are designed [42–45].

As suggested by the reviewer, in order to compare our approach with existing solutions and to determine whether using the whole gene regulatory structure outperforms solutions based on single regulatory regions, we trained and optimized 6 additional generative models using the same procedures as for ExpressionGAN (Methods M2,4). In addition to using whole single regions in the respective ranges as defined in Figure 1A, specifically the promoter (400 bp), UTRs (100 bp and 250 bp, respectively) and terminator (250 bp), we also used two shorter parts of the promoter featured in recent studies [6,8]. This included an 80 bp proximal promoter region located between -170 and -90 bp upstream of the transcription start site (TSS) [6,29] and the core promoter region located -170 bp upstream up to the TSS [8,46]. Firstly, we compared the dynamic ranges between either median or extreme predicted expression levels in generated sequence samples after 100,000 optimizer iterations of maximization and minimization (Figure 4A,B). Out of the single region generators, the terminator-based generator showed the highest expression range of ~3 orders of magnitude, whereas the 5' UTR and 80 bp proximal promoter-based generators resulted in the lowest ranges of ~1 and ~2 orders of magnitude, respectively. The dynamic range of ExpressionGAN was from 29% to 277% larger in the case of median expression values with best performing (terminator) and worst performing (5' UTR) generator variants,

respectively (Figure 4B), reflecting a 6 to 358-fold increase between median expression levels of maximization- and minimization-based sequence samples (Figure 4A).

Additionally, we have also performed an analysis of the relevance of different regulatory region combinations, adding the results to the supplementary information (Figure S7). Here, combinations of regions were occluded and absolute relevance scores were computed using the training dataset (see Methods M1,3). As expected, a larger amount of jointly occluded regions generally has a higher effect on perturbing gene expression levels, which is reflected in the absolute relevance score, with the largest effect observed when occluding combinations of three regions (Figure S7). Specifically, median absolute relevance values were 0.13, 0.19 and 0.24 with a number of 1, 2 or 3 occluded regions, respectively. Apart from the knowledge that promoter and UTRs are the key regulators of gene expression, we find that also the terminator region can enable the design of a relatively large dynamic range of expression levels compared to the other tested regions (Figure 4A,B). Unsurprisingly, it is therefore the combination of all regulatory regions that lead to the highest dynamic range, supporting the knowledge that the whole gene regulatory structure is important for fine-tuning gene expression [13,14]. We have also further discussed and clarified these aspects in an additional paragraph in the discussion section (lines 578-614, pages 22-23).

Finally, in order to further compare our approach with current state of the art solutions for regulatory DNA design [6,8], we used two of the above developed generative solutions, namely, the 80 bp proximal-promoter region [6] and 5' UTR region [9] (Figure 4). This enabled us to further verify the predictive capacity of our models, achieving significant correlation with published gene expression values (Figure 4C,E: Spearman's ρ was 0.51 and 0.55, p -value $< 1e-16$, respectively). Importantly, since we found that using whole gene regulatory structures leads to much larger dynamic ranges of gene expression, we tested how expanding the single regulatory parts to whole gene regulatory structure would affect their performance (i.e. expand their dynamic range). For both single-region generators, we sampled 128 of the existing sequence designs [6] and expanded them with all 4238 available native gene regulatory structures, yielding $2 \times 542,464$ sequence constructs that were used to analyze any additional dynamic potential with the already optimized short sequences. Indeed, we observed that a dynamic range spanning an order of magnitude of predicted expression levels was achievable with both solutions (Figure 4D,F: between 10th and 90th percentiles of expression levels). This suggests that short single-region sequence designs are not capable of controlling gene expression across its full dynamic range, despite the sequences being optimized previously [6,9]. In order to fully drive gene expression to its actual extremes, proper optimization of the gene regulatory structure with all adjacent regulatory regions is required.

The results described above are presented as a new chapter 4 in the results section (lines 271-337, pages 11-13) with a newly added Figure 4 that includes 6 panels and supplementary Figure S7. We have also expanded and revised the discussion section to clearly explain these results and put them into context of the whole study (lines 578-614, pages 22-23). Apart from this, we would like to restate that no other study has performed experimental validations of a fully optimized generative strategy to design regulatory DNA across a range of common expression levels (Figures 3 and 6A-D, please see reply 1.4 above), as we have done here.

Furthermore, we have also compared our method with another state of the art DNA design strategy - a mutational approach (Figure 6E-G, results lines 479-537, pages 18-20). Briefly, based on the knowledge that mutagenesis relies on testing arbitrary sequence mutations that can lead to predictor pathologies (please see Reply 1.5 and the revised Introduction section, lines 41-124, pages 3-5) we accurately controlled the amount of mutations in the designed candidates, producing only 1, 2, 5 and 10% of mutated sequence. In this way, we could control predictor functionality and potentially avoid pathologies, whereby generating sequences far outside the biologically allowed sequence space could push it into unpredictable behavior. Therefore, we computed a whole distribution of mutated sequences (Figure S20: altogether 16.8 million mutations tested) with different levels of mutations and expression levels, where we could controllably check which ones achieved desired changes. By testing the best sequence designs experimentally, we found that only those with 5 and 10% of mutations led to sequences that achieved the desired 2-fold expression level changes, and these were experimentally tested. However, despite sequences predicted to achieve >2-fold increase in expression level, they did not achieve a meaningful increase in expression levels compared to natural controls (Figure 6F, Table S4). Here, the key differentiator in the tested approaches was the type of sequence design approach that is joined with the predictive model: (i) either a generative approach that actually models the sequence-function landscape and provides the predictive screening model with valid candidates, or (ii) a 'blind' approach, that performs random mutagenesis and relies completely on predictor guidance, with random chance dictating whereas the predictor will produce accurate predictions. Due to predictor pathologies, the mutational approach restricts us to remain closeby in the tested sequence space in order to not produce too many infeasible sequences, whereas with the generative approach, we can traverse much further in the amount of sequence deviations from the training data (Figure 6A: <70% sequence similarity) and still get functional sequences. Therefore, our proof of concept results suggest that sticking 'smart' generation approaches to selection procedures can improve them, outperforming 'blind' design approaches.

Apart from the above stated manuscript changes, to increase the clarity of the results described in the above paragraph, we have performed overall proofreading and minor revisions to the text in chapter 6 of the results section (lines 433-537, pages 17-20) as well as completely revision of the third paragraph of that chapter as well as the text following it (lines 479-516, pages 18-19), in line with the revision of the introduction section (lines 41-124, pages 3-5, see Reply 1.2). We have also completely revised the paragraph in the discussion section describing this analysis (lines 648-682, page 24-25).

We hope that our revisions and additional set of analyses brings previously missed transparency and further clarifies our work.

Comment 1.6

How is the knowledge gained any different than what others have proposed previously? These overall metrics and comparisons are critical for understanding the impact of this work.

Reply 1.6

The key novelties in the present study include:

- (i) Learning the regulatory landscape of gene expression control directly from endogenous genomic and RNA-Seq data without performing experimental screening of random DNA;
- (ii) building generative models spanning the whole gene regulatory structure and not merely single regulatory regions (Figure 1A, please see Replies 1.2 and 1.3);
- (iii) devising an optimization strategy that connects the noncoding regulatory DNA properties with those of the coding regions to truly characterize the whole gene regulatory structure and enable traversal and targeted design of DNA across the whole dynamic range of gene expression levels (Figure 3);
- (iv) analysis of the capability of the generative models to recapitulate the natural DNA cis-regulatory grammar that is known to upregulate and downregulate gene expression levels [13,14,33–36] (Figures 2 and 5, please see Reply 1.3);
- (v) comparison of the use of whole gene regulatory structures and single regulatory regions or regulatory part as are commonly used in other approaches [6,8–10], showing that a large range of expression control remains potentially untapped with existing single-region approaches and whole gene regulatory structures offers a promising avenue for unlocking the full potential of gene expression control (Figure 4, please see Reply 1.5);
- (vi) *in vivo* experimental validation of our optimized ExpressionGAN approach across the most relevant range of gene expression levels spanning 3 orders of magnitude (Figure 6D) and preferentially selecting sequences with the lowest sequence similarity to natural ones and amongst themselves (Figure 6A,B, please see Reply 1.4);
- (vii) comparing our generative strategy with a mutational one to show how a generative approach modeling the sequence-function landscape directly designs biologically feasible sequences, which enables it to deviate much farther from the known sequence space and generate completely novel DNA designs with natural level of sequence divergence (Figure 6, please see Reply 1.5).

To facilitate improved and expanded content, transparency, clarity and placement of the present study in the state of the art of the present field, we have performed the following major revisions:

- complete revision of the abstract section (lines 24-36, page 2);
- complete expansion and rewriting of 2 paragraphs in the introduction section (lines 41-124, pages 3-5);
- in the results section, addition of chapter 4 (lines 271-337, pages 11-13) and apart from revision of text across all chapters for clarity, revision of chapter content to facilitate clarity, with manuscript now including 6 chapters instead of 4, revision and expansion of the first chapter (lines 127-182, pages 6-7), revision of the final chapter (lines 433-537, pages 17-20), ;

- in figures and tables, revision of 4 plots in figure 1, addition of Figure 4 with 6 plots (lines 322-337, page 13), addition of supplementary figures S7, S9 and table S5, revision of figure and table order in supplementary material;
- in the discussion section, apart from the whole text being revised, updated and expanded, a new paragraph was added (lines 578-614, pages 22-23) and another completely revised (lines 648-682, page 24-25);
- methods section updated;
- to improve the language and clarity, 3 separate native or well trained English speaking researchers have read and commented on the whole manuscript;
- references expanded with addition of 30 new references;
- the code and data have been made available over public repositories Github and Zenodo (see statements added at end, lines 932-942, page 34)

Reviewer #2 (Remarks to the Author):

Comment 2.1

In this work, Zrimec et al. introduce a method for designing regulatory non-coding DNA sequences in yeast using deep neural networks. Specifically, the authors train a Generative Adversarial Network (GAN) on published genomic and RNA-seq data with the purpose of generating novel regulatory sequences that contain endogenous-like features. The GAN learns to simultaneously generate promoter, UTR, and terminator sequences. Furthermore, they combine the GAN's generator network with a previously developed predictive model of gene expression to design sequences with specified performances. Finally, they experimentally test a small subset of designed promoters and find multiple sequences that result in stronger expression compared to highly expressed natural controls.

Overall, the work presented is a positive contribution to the synthetic biology literature, where deep neural networks trained on large publicly available datasets are increasingly being used for sequence design. It is noteworthy that the GAN can be successfully trained on endogenous data only. However, given that the task is to design endogenous-like sequences, perhaps this result is not that surprising. Furthermore, the authors do a comprehensive job at ensuring that the designed sequences contain native-like features while being different from endogenous sequences and from one another. Finally, the approach of connecting the GAN's generator to a predictor and optimizing in the latent space of the generator is simple yet effective.

Reply 2.1

We thank the reviewer for his positive and constructive comments, which we have addressed in full in the revised manuscript, with answers and explanations provided below.

Briefly, to facilitate improved and expanded content, transparency, clarity and placement of the present study in the state of the art of the present field, we have performed the following major revisions:

- complete revision of the abstract section (lines 24-36, page 2);
- complete expansion and rewriting of 2 paragraphs in the introduction section (lines 41-124, pages 3-5);
- in the results section, addition of chapter 4 (lines 271-337, pages 11-13) and apart from revision of text across all chapters for clarity, revision of chapter content to facilitate clarity, with manuscript now including 6 chapters instead of 4, revision and expansion of the first chapter (lines 127-182, pages 6-7), revision of the final chapter (lines 433-537, pages 17-20), ;

- in figures and tables, revision of 4 plots in figure 1, addition of Figure 4 with 6 plots (lines 322-337, page 13), addition of supplementary figures S7, S9 and table S5, revision of figure and table order in supplementary material;
- in the discussion section, apart from the whole text being revised, updated and expanded, a new paragraph was added (lines 578-614, pages 22-23) and another completely revised (lines 648-682, page 24-25);
- methods section updated;
- to improve the language and clarity, 3 separate native or well trained English speaking researchers have read and commented on the whole manuscript;
- references expanded with addition of 30 new references;
- the code and data have been made available over public repositories Github and Zenodo (see statements added at end, lines 932-942, page 34)

Comment 2.2

However, there are a few issues that I think need to be addressed before publication.

- The introduction frames the present study as an improvement over the traditional method of “experimental screening of large amounts of random synthetic sequences” (lines 64-65). This is written as to imply that most sequence design efforts have consisted of little more than testing fully random sequences chosen without any direction. In reality, the first decade of synthetic biology was dominated by the moderately successful rational design approach, where regulatory sequences were handcrafted and tested in low-throughput assays. A bit further on the authors also write (line 114-115): “we aimed to improve upon random mutagenesis.” In reality, there is a rich literature of papers that use generative models to design functional sequences. I strongly recommend the authors modify their introduction accordingly and add relevant references.

Reply 2.2

Yes, we are aware that painstaking sequence design and step-by-step design-and-test approaches were applied in the past to design regulatory DNA and uncover the basic principles of regulator functionality. We are thus not trying to imply that most sequence design efforts have consisted of little more than testing fully random sequences chosen without any direction. We have thus also specifically stated these types of design strategies in the introduction section (lines 59-62, page 3), building and testing regulators by stacking different known functional sequence motifs, which references recent efforts to design batches of minimal regulators by including well known DNA motifs at different positions, orientations and potentially multiples [36,43,47,48]. However, due to the amount of material that needs to be covered to give a proper background, whereas standard scientific paper-shaping directions limit the introduction to be short and concise, e.g. [2,3,6,29], we have made effort to shape the present introduction into a succinct overview that focuses on the points central to the present study and which are the key differentiators between the present approach and the most recent other approaches - including generative and mutagenesis studies.

As suggested by the reviewer, we have reshaped and expanded the introduction section (lines 41-124, pages 3-5), adding additional overview of recent approaches both from the mutational and generative fields. The key focus was to pinpoint and clearly explain the limitations of both types of approaches and how the present study goes beyond these limitations, trying to resolve them. With mutagenesis, this includes producing and exploring sequences with arbitrary random mutations in all the related approaches including genetic algorithms [1,2]. Since these sequence candidates can be biologically infeasible, it leads to untrustworthy predictions and resource inefficiency with multiple testing and design rounds [2-5] (lines 59-78, pages 3-4). Please see also Reply 2.5 below for a more detailed discussion.

Indeed, the literature of papers that use generative models to design functional sequences is rich, however focusing on design of regulatory DNA, there are actually not that many studies and they display big drawbacks, namely: (i) either without optimization [10], (ii) or on single regions (all of them), (iii) or not testing experimentally, testing e.g. polyadenylation but not transcription capability specifically [3], (iv) or genes but not regulatory regions [12] (lines 80-101, page 4). By including whole gene regulatory structure and experimentally verifying a portion of the design approach, we expand current results and provide a highly useful advancement in the field.

Comment 2.3

- It is very difficult to understand which regulatory sequence regions the generator model is actually trained on without reading the methods section. For example, lines 122 and 123 contain the vague descriptor “whole gene regulatory structures”, which I recommend should be replaced with the actual sequence regions (promoter, UTRs, terminator). Another set of vague descriptors are “variables describing the coding region” (lines 202-203) and “coding region information” (line 226), which should be replaced by “codon frequencies” if that is the only feature. Furthermore, Figure 1A needs to be fully reworked so that it clearly indicates this information. The current thin dashed lines below the regulatory regions are hard to notice, and the figure gives the overall impression that even CDS sequences are generated.

Reply 2.3

Figure 1A has been fixed as suggested by the reviewer. Namely, figure partitioning has been made more clear, with dashed lines removed where specified, and additional lines and text added to explain how data partitions were used for training either the predictors or generators, respectively. The text 'codon frequencies' was clearly written in a non-shortened form to make it clear this is what we are using. Otherwise the current version of the figure contains all the information and regulatory region limits required to assist the reader in understanding the specific part of our approach.

Additionally, to improve clarity, we have revised the introduction and expanded the first chapter of the results section (lines 127-182, pages 6-7), with this chapter split into two chapters (chapter 2 lines 182-221, pages 8-9). Accordingly, we have moved panels from the supplementary information to the main Figure 1B,C to visualize data properties and predictor performance. Figure 1D was similarly moved here to explain the generative adversarial network approach, as this is a key methodology in the study.

All of the reviewers' comments have been fixed as suggested, giving exact regions for the whole gene regulatory structure as well as codon frequencies (line 240, page 10). Due to this journal and manuscript being targeted for a wide audience, we have left the term “variables describing the coding region” (lines 239-240, page 10) as next to it is a clear description of the meaning that 64 codon frequencies are used, thus enabling both expert and general audience readers to understand this.

Comment 2.4

- There is a lack of baselines against which the presented method is evaluated.

Reply 2.4

The ExpressionGAN devised in the present study is based on existing thoroughly validated published methods, which include the following. (i) Generative adversarial networks (GAN) [15], which have been demonstrated as the top generative modeling approach for producing optimal sequence-function mappings [3,10,11,16,17]. (ii) An activation-maximization optimization [2,18–20] approach, which, due to multiple tests in the vision domain with human performance verification, has had its performance tested at a level equal to experimental validation [5,19,20]. (iii) Sequences spanning the whole gene regulatory structure [14] (Figure 1A) and (iv) highly-accurate predictive models of gene expression levels that can explain over 82% of expression variation from regulatory sequence alone (Figure 1B), both tested in a previous study [13]. This is detailed in the second paragraph of the discussion section (lines 550-576, pages 21-22).

We further test and validate this generative approach in numerous ways in separate controlled experiments, demonstrating its performance and usefulness, including the following comparisons. (i) Comparing the properties of generated sequences to natural ones (Figure 2). (ii) We expanded the results section (lines 271-337, pages 11-13) with additional analyses comparing the use of the whole gene regulatory structure to single regulatory regions and shorter promoter parts, which are commonly used with both generative approaches [10] and mutagenesis [6,8,9] (Figure 4). (iii) Testing whether generated sequences contain known DNA regulatory grammar properties that drive gene expression to low and high values (Figure 5). (iv) Experimental demonstration and validation of ExpressionGAN-generated sequences (Figure 6A-D), designed specifically to test as different sequences as possible (both to natural and among themselves, Figure 6A,B) to properly validate the approach. (v) Further experimentally contrasting the generative approach with a standard mutational one that does not inherently model the allowed sequence-function landscape (Figure 6E-F). The above comparisons are further explained and discussed in the third and following paragraphs of the discussion section (lines 578-682, pages 22-25).

Comment 2.5

It is not clear why the authors did not compare to approaches other than random mutagenesis such as genetic algorithms proper and in particular simulated annealing. In addition, while we don't expect the authors to run a full comparison against every model-based or generative neural network approach ever developed, they should more clearly acknowledge the existence of that literature.

Reply 2.5

We thank the reviewer for pointing this out. Indeed, we see that in the previous version of the manuscript we did not explain our rationale for the experiments and comparisons correctly nor clearly. We have now strived to improve this, focusing on improving all parts related to the specific topic of comparing mutational and generative strategies, in the respective introduction, results and discussion sections.

Specifically, we have expanded the literature overview in the present version, explaining the key limitations of mutational strategies in the light of generative approaches already in the introduction section (lines 41-124, pages 3-5). Notably, genetic algorithms and their derivatives also rely on random mutagenesis, introducing arbitrary mutations in every round of candidate generation, since they do not explicitly model the allowed sequence-function landscape in the sequence design step [1,6–9]. Rather, they rely fully on the computational screening step (by predictive models, also termed oracles) to understand sequence functionality and to guide sequence design [1,2]. However, predictive models are highly sensitive to sequence validity and displaying worsening performance with sequence deviations from the training data, especially when traversing towards biologically infeasible sequence subspaces. Since also infeasible sequences are produced in the random mutagenesis step, testing such candidates can stress the computational screening methods and lead them into pathological states, where predictions are highly untrustworthy [3–5]. Apart from this, only a small subset of sequences actually conform to the target design principles, and finding them is hard among the enormous sequence space, with many local minima that these algorithms can get stuck in [2]. Therefore, present mutational approaches are highly resource inefficient, as they require many separate runs to explore different settings, generally requiring unnecessary testing of many infeasible sequences as well as experiments to verify the functionality of the designed sequence sets. On the other hand, due to being capable of modeling the natural sequence-function landscape, generative approaches can directly draw biologically feasible sequence samples.

In our experiments, detailed in chapter 6 of the results section (lines 433-537, pages 17-20), we presumed that full-scale mutagenesis was unnecessary as we did not require testing the full dynamic range. Also, we could computationally afford a brute force mutational approach. This was highly advantageous, as, importantly, based on the knowledge that mutagenesis relies on testing arbitrary sequence mutations that can lead to predictor pathologies, we wanted to accurately control the amount of mutations in the designed candidates, producing only 1, 2, 5 and 10% of mutated sequence. In this way, we could control predictor functionality and potentially avoid pathologies, whereby generating sequences far outside the biologically allowed sequence space could push it into unpredictable behavior.

Therefore, instead of obtaining sequences from merely small specific subspaces, the most straightforward approach was to obtain a whole distribution of mutated sequences (Figure S20: altogether 16.8 million mutations tested) with different levels of mutations and expression levels, where we could controllably check which ones achieved desired changes. We thus produced sequences with desired levels just as we would with a potentially more advanced procedure, like genetic algorithms, just that we had the entire set of these candidates available for inspection. Of the sequence designs, only those with 5 and 10% of mutations led to sequences that achieved the desired 2-fold expression level changes, and these were experimentally tested. However, despite sequences predicted to achieve >2-fold increase in expression level, when tested experimentally, they did not achieve a meaningful increase in expression levels compared to natural controls (Figure 6F, Table S4).

On the other hand, as detailed in the same chapter of results (lines 433-537, pages 17-20), by testing sequence designs generated by the generative models, we found that sequences could achieve the overall predicted expression levels (Figure 6D, Figure S17, Spearman's $\rho = 0.74$, p -value $< 1.6e-14$) and 57% of them surpassed the naturally highly-expressed controls. This was achieved despite preferentially selecting sequences with the lowest sequence similarity to natural ones (Figure 6A, Figure S14: >30% difference in avg. sequence identity) and maximizing sequence diversity in each expression range (Figure 6B, Figure S14), meaning that the sequences deviated much farther from any known natural variant compared to those tested with the mutational procedure. Despite this, the sequences were feasible and properly recognized by the predictor (Spearman's $\rho = 0.74$, p -value $< 1.6e-14$).

This key differentiator in the tested approaches is thus not in the type of mutational approach (i.e. either brute force randomization or iterating via genetic selections), as all of them rely on random mutagenesis, but in the type of sequence design approach that is joined with the predictor: (i) either a generative approach that actually models the sequence-function landscape and provides the predictive screening model with valid candidates, or (ii) a 'blind' approach, that performs random mutagenesis and relies completely on predictor guidance, with random chance dictating whereas the predictor will produce accurate predictions. Due to predictor pathologies, the mutational approach restricts us to remain closeby in the tested sequence space in order to not produce too many infeasible sequences, whereas with the generative approach, we can traverse much further in the amount of sequence deviations from the training data (Figure 6A: <70% sequence similarity) and still get functional sequences. Therefore, our proof of concept results suggest that sticking 'smart' generation approaches to selection procedures can improve them, outperforming 'blind' design approaches.

Apart from the above stated manuscript changes, to increase the clarity of the results described in the above paragraph, we have performed overall proofreading and minor revisions to the text in chapter 6 of the results section as well as completely revision of the third paragraph of that chapter as well as the text following it (lines 479-516, pages 18-19), in line with the revision of the introduction section (lines 41-124, pages 3-5, please see Reply 2.2). We have also completely revised the paragraphs in the discussion section describing this analysis (lines 578-614, pages 22-23, and lines 648-682, pages 24-25).

Comment 2.6

- The paper spends a lot of time dissecting the features of the generated sequences within each individual regulatory element, with some interesting + detailed analysis. It would have been interesting to see some kind of ablation study here, e.g. training a generator on the leave-one-out permutations of the promoter, 5'-UTR, 3'-UTR, terminator. This would make a more convincing argument on whether using the entire regulatory ecosystem for deep learning-based sequence design offers a real advantage; and would also lend the individual regulatory analysis breakdown more impact. Alternatively, it might suggest e.g. some regulatory regions have higher impact or are easier targets for design/have wider dynamic range, which would also be interesting.

Reply 2.6

Yes, we agree with the reviewer that it is beneficial to apply a rational engineering design aspect, as it can be very beneficial to the study overall. We left this out due to our knowledge from past research as well as the focus and large amount of work with all the development and testing of other key issues included in the study (please see reply 2.4 giving a general overview of all the analyses and comparisons performed). Already in previous studies, including our own recent one [13], it was shown that regulatory regions each contribute a certain amount of information for predicting gene expression levels [9,28,37–40], with each region carrying a given amount of overlapping as well as unique information, thus jointly contributing to gene expression variation [13,14]. In fact, whereas the initiating 5' regions (promoters and 5' UTRs) seem to define large scale gene expression properties (turning expression on/off), the role of terminating 3' regions (including 3' UTRs and terminators) is to fine-tune expression levels [14,41]. Hence, varying a regulatory region while keeping others intact can lead to a large measurable degree of regulatory freedom per gene [13]. Additionally, from the technical point of view, only the whole gene regulatory structure constrains the functional sequence space to the biologically feasible regulatory variants. Even when sampling only a single region, it must still be based on the knowledge of the effect of the whole surrounding regulatory structure, as this is the primary state of natural regulatory systems [13,14,41] according to which also experimental systems are designed [42–45].

As suggested by the reviewer, we have now included a whole new chapter 4 in the results section (lines 271-337, pages 11-13), comparing the use of whole gene regulatory structures with solutions based on single regions and smaller parts that are commonly used in other studies [6,8–10] (Figure 4, please see Reply 2.7 for more details). Moreover, we have also performed an analysis of the relevance of different regulatory region combinations, adding the results to the supplementary information (Figure S7). Here, combinations of regions were occluded and absolute relevance scores were computed using the training dataset (see Methods M1,3). As expected, a larger amount of jointly occluded regions generally has a higher effect on perturbing gene expression levels, which is reflected in the absolute relevance score, with the largest effect observed when occluding combinations of three regions (Figure S7). Specifically, median absolute relevance values were 0.13, 0.19 and 0.24

with a number of 1, 2 or 3 occluded regions, respectively. Apart from the knowledge that promoter and UTRs are the key regulators of gene expression, we find that also the terminator region can enable the design of a relatively large dynamic range of expression levels compared to the other tested regions (Figure 4A,B). Unsurprisingly, it is therefore the combination of all regulatory regions that lead to the highest dynamic range, supporting the knowledge that the whole gene regulatory structure is important for fine-tuning gene expression [13,14]. We have also further discussed and clarified these aspects by expanding the discussion section with an additional paragraph (lines 578-614, pages 22-23).

Comment 2.7

- Closely related to the previous point: one of the purported advantages of the authors' design method is the ability to simultaneously design promoter, UTR, and terminator sequences to optimize expression. However, the extent to which this is an improvement over separately optimizing them is not explored. The following analysis could help address this issue: 1) Train separate GANs and predictors on each regulatory region. 2) Use these separate networks to design regulatory elements in isolation to maximize expression. 3) Use the combined predictive model to determine whether these elements in combination would lead to higher expression.

Reply 2.7

As suggested by the reviewer, in order to compare our approach with existing solutions and to determine whether using the whole gene regulatory structure outperforms solutions based on single regulatory regions, we trained and optimized 6 additional generative models using the same procedures as for ExpressionGAN (Methods M2,4). In addition to using whole single regions in the respective ranges as defined in Figure 1A, specifically the promoter (400 bp), UTRs (100 bp and 250 bp, respectively) and terminator (250 bp), we also used two shorter parts of the promoter featured in recent studies [6,8]. This included an 80 bp proximal promoter region located between -170 and -90 bp upstream of the transcription start site (TSS) [6,29] and the core promoter region located -170 bp upstream up to the TSS [8,46]. Firstly, as suggested by the reviewer already in Comment 2.6, we compared the dynamic ranges between either median or extreme predicted expression levels in generated sequence samples after 100,000 optimizer iterations of maximization and minimization (Figure 4A,B). Out of the single region generators, the terminator-based generator showed the highest expression range of ~3 orders of magnitude, whereas the 5' UTR and 80 bp proximal promoter-based generators resulted in the lowest ranges of ~1 and ~2 orders of magnitude, respectively. The dynamic range of ExpressionGAN was from 29% to 277% larger in the case of median expression values with best performing (terminator) and worst performing (5' UTR) generator variants, respectively (Figure 4B), reflecting a 6 to 358-fold increase between median expression levels of maximization- and minimization-based sequence samples (Figure 4A).

Additionally, in order to further compare our approach with current state of the art solutions for regulatory DNA design [6,8], we used two of the above developed generative solutions, namely, the 80 bp proximal-promoter region [6] and 5' UTR region [9] (Figure 4). This enabled us to further verify the predictive capacity of our models, achieving significant correlation with published gene expression values (Figure 4C,E: Spearman's ρ was 0.51 and 0.55, p -value $< 1e-16$, respectively). Importantly, since we found that using whole gene regulatory structures leads to much larger dynamic ranges of gene expression, we tested how expanding the single regulatory parts to whole gene regulatory structure would affect their performance (i.e. expand their dynamic range). For both single-region generators, we randomly sampled 128 of the existing sequence designs [6] and expanded them with all 4238 available native gene regulatory structures, yielding $2 \times 542,464$ sequence constructs that were used to analyze any additional dynamic potential with the already optimized short sequences. Indeed, we observed that a dynamic range spanning an order of magnitude of

predicted expression levels was achievable with both solutions (Figure 4D,F: between 10th and 90th percentiles of expression levels). This suggests that short single-region sequence designs might not be capable of controlling gene expression across its full dynamic range, despite the sequences being optimized in their restricted sense [6,9]. In order to fully drive gene expression to its actual extremes, proper optimization of the gene regulatory structure with all adjacent regulatory regions is required. The results are presented as a new chapter 4 in the results section (lines 271-337, pages 11-13) with a newly added Figure 4 that includes 6 panels and supplementary Figure S7.

Comment 2.8

- Please include the 24 designed sequences that could not be synthesized by either twist or IDT?

Reply 2.8

The generated sequences that could not be synthesized have been added as Table S5.

Comment 2.9

- The authors mention in several places that their method allows designing gene expression across 6 orders of magnitude (e.g. line 459), yet their experimental results show coverage of little more than two (Figure 5D). Please clarify.

Reply 2.9

Yes, with computational predictions, by comparing generated sequences with the lowest and highest predicted expression levels, we obtain a maximum dynamic range spanning almost 6 orders of magnitude (Figure 4A,B). The experiments show a much lower coverage than the predictions, however we were currently not able to test the method across a larger dynamic range. We reported on the possibilities based on our computational assessment of the method, as detailed in chapters 3 and 4 in the results section (lines 223-270, pages 10-11 and lines 271-337, pages 11-13, respectively, Figures 3B,C and 4A,B). To note, since many studies have not been thoroughly experimentally tested in the given field, reporting on computationally assessed capabilities is commonly done by many other studies. Nevertheless, we agree that it can be misleading to the readers to specify this value without specifying the way it was obtained, and we have thus corrected this statement in the discussion, by specifying that it was obtained according to computational predictions, with also those figures specified (line 566, page 21), whereas the experimental results linking to that figure are mentioned in the next sentence (line 569, page 21).

Comment 2.10

- Minor point: can the authors elaborate on why they find positive correlation between the number of adenines around the start codon and predicted expression levels of generated sequences (lines 261-262)? The authors seem to be making the connection to Kozak sequences (line 259). However, Kozak sequences regulate translation, but the training data for the models in this study is genomic and RNA-seq information.

Reply 2.10

Indeed, untranslated regions and the specific regulatory elements within generally affect translation. With 5' UTRs, the key factors for translation initiation are ribosome recruitment to the mRNA and correct positioning over the start codon, where the presence of a Kozak sequence [49] in the 5' UTR increases the efficiency of translation [24,50]. Here, also nucleotides upstream of Kozak were shown to be involved in transcriptional regulation [25]. On the other hand, the overall levels of mRNA expression, as measured by RNA-seq, are defined both by rates of synthesis and degradation (also termed mRNA stability). Despite not excluding possible effects on transcription [38], multiple studies have demonstrated a general coupling between translation and mRNA degradation [51–53] and suggest that 5' UTRs affect mRNA levels via translation-mediated RNA degradation [38,54–56]. The 5' UTRs and Kozak sequence context thus functionally affects mRNA levels by regulating mRNA stability [38,55,57], with different amounts of mRNA degradation observed based on weak or strong Kozak sequences [58]. We thank the reviewer for pointing this out and have thus expanded the text in the results section to clarify it (lines 361-363, page 14).

Comment 2.11

- The following figure references might be incorrect: Figure 2G (line 365) and Figure 2F (line 371).

Reply 2.11

We have fixed the figure references in question and checked all others for correctness.

References

1. Eiben AE, Smith J. From evolutionary computation to the evolution of things. *Nature*. 2015;521: 476–482.
2. Bogard N, Linder J, Rosenberg AB, Seelig G. A Deep Neural Network for Predicting and Engineering Alternative Polyadenylation. *Cell*. 2019;178: 91–106.e23.
3. Linder J, Bogard N, Rosenberg AB, Seelig G. A Generative Neural Network for Maximizing Fitness and Diversity of Synthetic DNA and Protein Sequences. *Cell Syst*. 2020;11: 49–62.e16.
4. Szegedy C, Zaremba W, Sutskever I, Bruna J, Erhan D, Goodfellow I, et al. Intriguing properties of neural networks. *arXiv [cs.CV]*. 2013. Available: <http://arxiv.org/abs/1312.6199>
5. Nguyen A, Yosinski J, Clune J. Deep Neural Networks are Easily Fooled: High Confidence Predictions for Unrecognizable Images. *arXiv [cs.CV]*. 2014. Available: <http://arxiv.org/abs/1412.1897>
6. Vaishnav ED, de Boer CG, Molinet J, Yassour M, Fan L, Adiconis X, et al. The evolution, evolvability and engineering of gene regulatory DNA. *Nature*. 2022;603: 455–463.
7. Sample PJ, Wang B, Reid DW, Presnyak V, McFadyen IJ, Morris DR, et al. Human 5' UTR design and variant effect prediction from a massively parallel translation assay. *Nat Biotechnol*. 2019;37: 803–809.
8. Jores T, Tonnie J, Wrightsman T, Buckler ES, Cuperus JT, Fields S, et al. Synthetic promoter designs enabled by a comprehensive analysis of plant core promoters. *Nat Plants*. 2021;7: 842–855.
9. Cuperus JT, Groves B, Kuchina A. Deep learning of the regulatory grammar of yeast 5' untranslated regions from 500,000 random sequences. *Genome Res*. 2017;27: 1–10.
10. Wang Y, Wang H, Wei L, Li S, Liu L, Wang X. Synthetic promoter design in *Escherichia coli* based on a deep generative network. *Nucleic Acids Res*. 2020;48: 6403–6412.
11. Killoran N, Lee LJ, DeLong A, Duvenaud D, Frey BJ. Generating and designing DNA with deep generative models. *arXiv [cs.LG]*. 2017. Available: <http://arxiv.org/abs/1712.06148>
12. Gupta A, Zou J. Feedback GAN for DNA optimizes protein functions. *Nature Machine Intelligence*. 2019;1: 105–111.
13. Zrimec J, Börlin CS, Buric F, Muhammad AS, Chen R, Siewers V, et al. Deep learning suggests that gene expression is encoded in all parts of a co-evolving interacting gene regulatory structure. *Nat Commun*. 2020;11: 6141.
14. Zrimec J, Buric F, Kokina M, Garcia V, Zelezniak A. Learning the regulatory code of gene expression. *Front Mol Biosci*. 2021;8. doi:10.3389/fmolb.2021.673363
15. Goodfellow IJ, Pouget-Abadie J, Mirza M, Xu B, Warde-Farley D, Ozair S, et al. Generative Adversarial Networks. *arXiv [stat.ML]*. 2014. Available: <http://arxiv.org/abs/1406.2661>
16. Repecka D, Jauniskis V, Karpus L, Rembeza E, Rokaitis I, Zrimec J, et al. Expanding

- functional protein sequence spaces using generative adversarial networks. *Nature Machine Intelligence*. 2021. doi:10.1038/s42256-021-00310-5
17. Riesselman AJ, Ingraham JB, Marks DS. Deep generative models of genetic variation capture the effects of mutations. *Nat Methods*. 2018;15: 816–822.
 18. Killoran N, Lee LJ, DeLong A, Duvenaud D, Frey BJ. Generating and designing DNA with deep generative models. *arXiv [cs.LG]*. 2017. Available: <http://arxiv.org/abs/1712.06148>
 19. Simonyan K, Vedaldi A, Zisserman A. Deep Inside Convolutional Networks: Visualising Image Classification Models and Saliency Maps. *arXiv [cs.CV]*. 2013. Available: <http://arxiv.org/abs/1312.6034>
 20. Yosinski J, Clune J, Nguyen A, Fuchs T, Lipson H. Understanding Neural Networks Through Deep Visualization. *arXiv [cs.CV]*. 2015. Available: <http://arxiv.org/abs/1506.06579>
 21. Inukai S, Kock KH, Bulyk ML. Transcription factor–DNA binding: beyond binding site motifs. *Curr Opin Genet Dev*. 2017;43: 110–119.
 22. Rhee HS, Pugh BF. Genome-wide structure and organization of eukaryotic pre-initiation complexes. *Nature*. 2012;483: 295–301.
 23. Yang C, Bolotin E, Jiang T, Sladek FM, Martinez E. Prevalence of the initiator over the TATA box in human and yeast genes and identification of DNA motifs enriched in human TATA-less core promoters. *Gene*. 2007;389: 52–65.
 24. Nakagawa S, Niimura Y, Gojobori T, Tanaka H, Miura K-I. Diversity of preferred nucleotide sequences around the translation initiation codon in eukaryote genomes. *Nucleic Acids Res*. 2008;36: 861–871.
 25. Li J, Liang Q, Song W, Marchisio MA. Nucleotides upstream of the Kozak sequence strongly influence gene expression in the yeast *S. cerevisiae*. *J Biol Eng*. 2017;11: 25.
 26. Guo Z, Sherman F. 3'-end-forming signals of yeast mRNA. *Trends Biochem Sci*. 1996;21: 477–481.
 27. Zhao J, Hyman L, Moore C. Formation of mRNA 3' ends in eukaryotes: mechanism, regulation, and interrelationships with other steps in mRNA synthesis. *Microbiol Mol Biol Rev*. 1999;63: 405–445.
 28. Shalem O, Sharon E, Lubliner S, Regev I, Lotan-Pompan M, Yakhini Z, et al. Systematic dissection of the sequence determinants of gene 3'end mediated expression control. *PLoS Genet*. 2015;11: e1005147.
 29. de Boer CG, Vaishnav ED, Sadeh R, Abeyta EL, Friedman N, Regev A. Deciphering eukaryotic gene-regulatory logic with 100 million random promoters. *Nat Biotechnol*. 2020;38: 56–65.
 30. Zhou T, Shen N, Yang L, Abe N, Horton J, Mann RS, et al. Quantitative modeling of transcription factor binding specificities using DNA shape. *Proc Natl Acad Sci U S A*. 2015;112: 4654–4659.
 31. Yang L, Orenstein Y, Jolma A, Yin Y, Taipale J, Shamir R, et al. Transcription factor family-specific DNA shape readout revealed by quantitative specificity models. *Mol Syst Biol*. 2017;13: 910.

32. Zrimec J. Structural representations of DNA regulatory substrates can enhance sequence-based algorithms by associating functional sequence variants. Proceedings of the 11th ACM International Conference on Bioinformatics, Computational Biology and Health Informatics. New York, NY, USA: Association for Computing Machinery; 2020. pp. 1–6.
33. Lubliner S, Keren L, Segal E. Sequence features of yeast and human core promoters that are predictive of maximal promoter activity. *Nucleic Acids Res.* 2013;41: 5569–5581.
34. Struhl K, Segal E. Determinants of nucleosome positioning. *Nat Struct Mol Biol.* 2013;20: 267–273.
35. Liu R, Liu L, Li X, Liu D, Yuan Y. Engineering yeast artificial core promoter with designated base motifs. *Microb Cell Fact.* 2020;19: 38.
36. Zürcher E, Tavor-Deslex D, Lituiev D, Enkerli K, Tarr PT, Müller B. A robust and sensitive synthetic sensor to monitor the transcriptional output of the cytokinin signaling network in planta. *Plant Physiol.* 2013;161: 1066–1075.
37. Espinar L, Schikora Tamarit MÀ, Domingo J, Carey LB. Promoter architecture determines cotranslational regulation of mRNA. *Genome Res.* 2018;28: 509–518.
38. Dvir S, Velten L, Sharon E, Zeevi D. Deciphering the rules by which 5'-UTR sequences affect protein expression in yeast. *Proc Natl Acad Sci.* 2013;110: E2792–E2801.
39. Cheng J, Maier KC, Avsec Ž, Rus P, Gagneur J. Cis-regulatory elements explain most of the mRNA stability variation across genes in yeast. *RNA.* 2017;23: 1648–1659.
40. Morse NJ, Gopal MR, Wagner JM, Alper HS. Yeast Terminator Function Can Be Modulated and Designed on the Basis of Predictions of Nucleosome Occupancy. *ACS Synth Biol.* 2017;6: 2086–2095.
41. Washburn JD, Mejia-Guerra MK, Ramstein G, Kremling KA, Valluru R, Buckler ES, et al. Evolutionarily informed deep learning methods for predicting relative transcript abundance from DNA sequence. *Proc Natl Acad Sci U S A.* 2019;116: 5542–5549.
42. Dhillon N, Shelansky R, Townshend B, Jain M, Boeger H, Endy D, et al. Permutational analysis of *Saccharomyces cerevisiae* regulatory elements. *Synth Biol.* 2020;5: ysaa007.
43. Cai Y-M, Kallam K, Tidd H, Gendarini G, Salzman A, Patron NJ. Rational design of minimal synthetic promoters for plants. *Nucleic Acids Res.* 2020;48: 11845–11856.
44. Keren L, Zackay O, Lotan-Pompan M, Barenholz U, Dekel E, Sasson V, et al. Promoters maintain their relative activity levels under different growth conditions. *Mol Syst Biol.* 2013;9: 701.
45. Yamanishi M, Ito Y, Kintaka R, Imamura C, Katahira S, Ikeuchi A, et al. A genome-wide activity assessment of terminator regions in *Saccharomyces cerevisiae* provides a "terminatome" toolbox. *ACS Synth Biol.* 2013;2: 337–347.
46. Lubliner S, Regev I, Lotan-Pompan M, Edelheit S, Weinberger A, Segal E. Core promoter sequence in yeast is a major determinant of expression level. *Genome Res.* 2015;25: 1008–1017.
47. Redden H, Alper HS. The development and characterization of synthetic minimal yeast

- promoters. *Nat Commun.* 2015;6: 7810.
48. Curran KA, Morse NJ, Markham KA, Wagman AM, Gupta A, Alper HS. Short Synthetic Terminators for Improved Heterologous Gene Expression in Yeast. *ACS Synth Biol.* 2015;4: 824–832.
 49. Kozak M. Point mutations define a sequence flanking the AUG initiator codon that modulates translation by eukaryotic ribosomes. *Cell.* 1986;44: 283–292.
 50. Hinnebusch AG, Ivanov IP, Sonenberg N. Translational control by 5'-untranslated regions of eukaryotic mRNAs. *Science.* 2016;352: 1413–1416.
 51. Roy B, Jacobson A. The intimate relationships of mRNA decay and translation. *Trends Genet.* 2013;29: 691–699.
 52. Huch S, Nissan T. Interrelations between translation and general mRNA degradation in yeast. *Wiley Interdiscip Rev RNA.* 2014;5: 747–763.
 53. Radhakrishnan A, Green R. Connections Underlying Translation and mRNA Stability. *J Mol Biol.* 2016;428: 3558–3564.
 54. Muhlrاد D, Parker R. Recognition of yeast mRNAs as “nonsense containing” leads to both inhibition of mRNA translation and mRNA degradation: implications for the control of mRNA decapping. *Mol Biol Cell.* 1999;10: 3971–3978.
 55. Schwartz David C., Parker Roy. Mutations in Translation Initiation Factors Lead to Increased Rates of Deadenylation and Decapping of mRNAs in *Saccharomyces cerevisiae*. *Mol Cell Biol.* 1999;19: 5247–5256.
 56. Barnes CA. Upf1 and Upf2 proteins mediate normal yeast mRNA degradation when translation initiation is limited. *Nucleic Acids Res.* 1998;26: 2433–2441.
 57. LaGrandeur T, Parker R. The cis acting sequences responsible for the differential decay of the unstable MFA2 and stable PGK1 transcripts in yeast include the context of the translational start codon. *RNA.* 1999;5: 420–433.
 58. Acevedo JM, Hoermann B, Schlimbach T, Teleman AA. Changes in global translation elongation or initiation rates shape the proteome via the Kozak sequence. *Sci Rep.* 2018;8: 4018.

Reviewers' Comments:

Reviewer #1:

Remarks to the Author:

The authors have indeed addressed all of my prior concerns and have greatly improved the impact and readability of this manuscript. This is an interesting study that will lead to others using similar GAN approaches to broad datasets.

Reviewer #2:

Remarks to the Author:

The paper improved on revision and can be published.

I found the reply to the reviewers difficult to follow. Many pages of text are used to argue points that only the reviewers will see. I would prefer it if the authors focused on highlighting the changes they made /in the paper/ in response to the concerns raised.